# ECNet is an evolutionary context-integrated deep learning framework for protein engineering

Yunan Luo [1,3], Guangde Jiang[2,3], Tianhao Yu[2], Yang Liu[1], Lam Vo[2], Hantian Ding[1], Yufeng Su[1], Wesley Wei Qian[1], Huimin Zhao [2✉] & Jian Peng [1✉]

Machine learning has been increasingly used for protein engineering. However, because the general sequence contexts they capture are not specific to the protein being engineered, the accuracy of existing machine learning algorithms is rather limited. Here, we report ECNet (evolutionary context-integrated neural network), a deep-learning algorithm that exploits evolutionary contexts to predict functional fitness for protein engineering. This algorithm integrates local evolutionary context from homologous sequences that explicitly model residue-residue epistasis for the protein of interest with the global evolutionary context that encodes rich semantic and structural features from the enormous protein sequence universe. As such, it enables accurate mapping from sequence to function and provides generalization from low-order mutants to higher-order mutants. We show that ECNet predicts the sequence-function relationship more accurately as compared to existing machine learning algorithms by using ~50 deep mutational scanning and random mutagenesis datasets. Moreover, we used ECNet to guide the engineering of TEM-1 β-lactamase and identified variants with improved ampicillin resistance with high success rates.

[1] Department of Computer Science, University of Illinois at Urbana-Champaign, Urbana-Champaign, IL, USA. [2] Department of Chemical and Biomolecular Engineering, University of Illinois at Urbana-Champaign, Urbana-Champaign, IL, USA. [3]These authors contributed equally: Yunan Luo, Guangde Jiang. ✉email: zhao5@illinois.edu; jianpeng@illinois.edu

Protein engineering aims to create protein variants with improved or novel functions. One powerful protein engineering strategy is directed evolution, which consists of iterative cycles of mutagenesis and high-throughput screening or selection[1–3]. While directed evolution is highly successful, the protein sequence space that can be sampled by directed evolution is limited and developing an effective high-throughput screening or selection can require a significant experimental effort[4].

To address these limitations, machine learning (ML) algorithms have been developed to assist directed evolution, which led to many successfully engineered proteins[4–9]. In ML-assisted directed evolution, a machine learning model is trained to learn the sequence-function relationship from sequence and screening data. In one round of directed evolution, the model simulates and predicts the fitness of all possible sequences, and a restricted list of best-performing variants is used as the starting point for the next round of directed evolution. In contrast to the classical directed evolution, ML-assisted directed evolution can escape from the local optimum by learning the entire functional landscape from data. It takes full advantage of all available sequence and screening data, including those of unimproved variants, thereby traversing the fitness landscape more efficiently.

A critical component of ML-guided directed evolution is to build a machine learning algorithm that accurately maps sequence to function. Unlike the qualitative predictions that group protein sequences into different functional classes[10–13], in protein engineering, a model is required to distinguish quantitative functional levels of closely related sequences. For example, in one round of directed evolution, the ML model needs to predict the fitness of a sequence that differs from the parent sequence by only one or very few single amino acids. Several ML algorithms have been developed to predict the mutational effects by leveraging the evolutionary information of homologous sequences[14,15]. These methods built generative models to reveal the underlying constraints of the evolutionary process, which can then be used to infer which mutations are more tolerable or favorable than others. Because of the unsupervised nature, however, these methods are not able to leverage the fitness data of tested variants available during the directed evolution process and thus may have limited accuracy when guiding the protein engineering. More recently, inspired by the advances in natural language processing[16], an emerging trend is to pre-train a language model (LM) on large protein sequence datasets to learn the distribution of protein sequences[13,17–23]. The protein sequences observed in nature today are the results of natural selection by evolution. Out of the possible mutations to a sequence, evolution samples those that preserve or improve the protein's fitness, such as stability, structure, and function. The underlying constraints or factors that determine protein's fitness have shaped the distribution of protein sequences. LMs are used to unravel the 'grammars' or 'semantics' of sequence generation by evolution. By being trained on natural sequences to predict the likelihood that a particular amino acid appears within a context, the language model learns representations that are semantically rich and encode structure, evolutionary and biophysical contexts[17]. Several recent studies found that the representations learned by LMs can be used to predict the sequence-function relationship in an unsupervised way[24–26]. It was also found that using the learned representation as the feature input to fine-tune a supervised model improves fitness prediction on multiple protein mutagenesis datasets[18]. However, as these models are trained on massive sequences such as those in UniProt[27] and Pfam[28], the learned representations only capture general context for a wide spectrum of proteins but may not be specific to the protein to be engineered. Lacking this specificity in the representation, the prediction model may not be effective in capturing the underlying mechanism (e.g., epistasis between residues) that determines the fitness of a protein and is not able to effectively prioritize best-performing variants to assist the directed evolution.

In this work, we developed ECNet (evolutionary context-integrated neural network), a deep learning model that guides protein engineering by predicting protein fitness from the sequence. We constructed a sequence representation that incorporated the local evolutionary context specific to the protein to be engineered. This representation explicitly encodes the residue interdependencies of all residue pairs in the sequence, which informs our prediction model to quantify the effects of mutations —especially higher-order mutations—in the sequence. We further incorporated global evolutionary context from an LM model trained on large sequence databases to model the semantic grammar within protein sequences as well as other structure and stability relevant contexts. Finally, a recurrent neural network model, trained on the fitness data of screened variants, is used for the sequence-to-function modeling with both representations. Through extensive benchmarking experiments, we showed that ECNet outperforms existing methods on ~50 deep mutagenesis datasets. Further experiments on combinatorial mutagenesis datasets demonstrated that ECNet enables generalization from low-order mutants to higher-order mutants. Moreover, ECNet was successfully used to engineer TEM-1 β-lactamase variants with improved resistance to ampicillin.

## Results

**Residue co-evolution correlates protein functional fitness**. Mutations within the protein sequence can affect fitness in a non-independent way, which is also known as genetic interactions or epistasis. It was found that epistasis interactions, quantified by deep mutational scanning (DMS) of proteins, can be used to infer protein contacts and structures[29,30]. As structurally proximal protein residues are often inferred from co-variation pairs from sequence evolution historically[31,32], we hypothesized that co-evolution information can also be used to infer epistasis or fitness of proteins.

To test this hypothesis, we investigated the relationship between the co-evolution of residue pairs and the fitness of double mutants. We collected a DMS study that measured the fitness of double mutants of the human YAP65 WW domain[33]. We also quantified the strength of pairwise residue dependencies by fitting a direct coupling analysis model[34] to the homologous sequences of the WW domain (see the "Methods" section). We found that the strength of pairwise dependencies correlated with the fitness of double mutants (Spearman correlation 0.35; Fig. 1a). Similar to a previous study[14], we also used the change of dependency strength (by contrasting the mutant sequence to the wild-type sequence) to predict the fitness of protein variants in a set of DMS studies[35]. We found that the predictions correlated with experimental data with a Spearman correlation ranging from 0.1 to 0.5 (Fig. 1b). In addition, we observed a trend of increasing correlation score if a protein has more homologous sequences, presumably because abundant homologous sequences lead to a more accurately fitted direct coupling analysis model. Overall, these results suggested that there are signals in the evolution information that we can leverage to predict protein fitness. This motivated us to integrate evolutionary information of protein sequences to empower a supervised model that predicts the fitness of protein variants in directed evolution.

**Sequence-to-function modeling**. We built a deep learning sequence-to-function model, ECNet, that learns the mapping from protein sequences to their respective functional measurements (Fig. 1c, Supplementary Fig. 1) from data (e.g., fitness

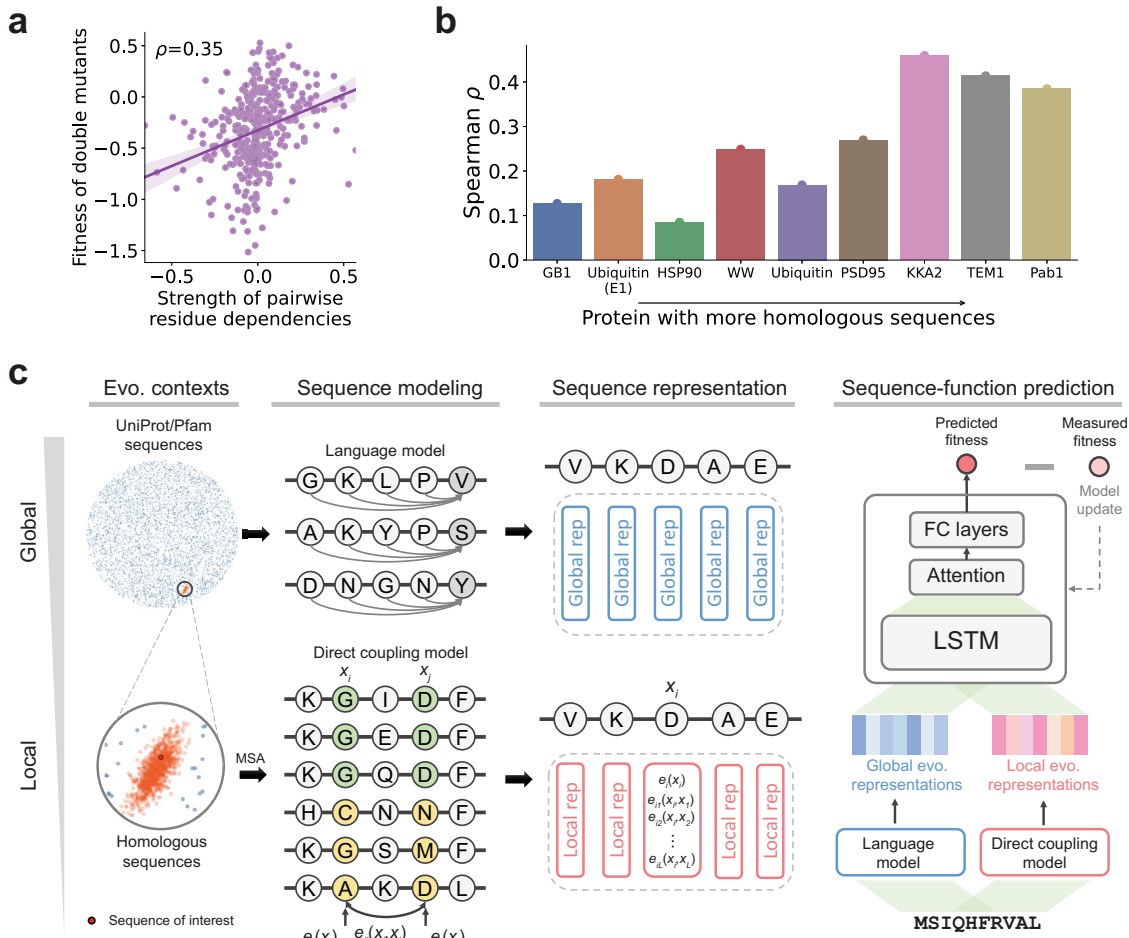

**Fig. 1 The motivation and overview of our evolutionary context-integrated sequence modeling method for protein engineering. a** Sequence co-evolution data correlates with fitness measurements in deep mutational scanning studies. The scatter plot shows the relationship between the fitness measurement of double mutants and the co-variation strength of residues where the mutations were introduced. Each data point represents a double mutant. The error band indicates the 95% confidence interval of the regression line. **b** Sequence co-evolution data can be used to predict protein fitness. The bar plot shows the Spearman correlation between experimentally measured fitness and strength changes of co-variation. Proteins were sorted by the number of homologous sequences. **c** An overview of ECNet, our evolutionary context-integrated deep learning framework for protein engineering. ECNet integrates global and local evolutionary contexts to represent the protein sequence of interest. First, a language model is used to learn global semantic-rich global sequence representations from the protein sequence databases such as UniProt or Pfam. Next, a direct coupling analysis model is used to capture the dependencies between residues in protein sequences, which encodes the local evolutionary context. The global and local evolutionary representations are then combined as sequence representations and used as the input of a deep learning model that predicts the fitness of proteins. Quantitative fitness data measured by deep mutational scanning (DMS) are used to supervise the training of the deep learning model (MSA: multiple sequence alignment; Dim. reduction: dimensionality reduction; LSTM: long short-term memory network; FC layers: fully-connected layers; Evo. contexts: evolutionary contexts; Evo. representations: evolutionary representations; Global/Local rep: global/local representation).

measured by deep mutational scanning). We used the LSTM neural network architecture and trained protein-specific models using large-scale deep mutational scanning datasets ("Methods").

Our model is mainly empowered by two informative protein representations, with one accounting for residue interdependencies of the specific protein of interest and the other capturing the general sequence semantics in the protein universe. Existing tools predict the conservation effects of mutations by considering each amino acid independently (e.g., PolyPhen-2[36] and CADD[37]) while others exploit structure information (e.g., FoldX[38] and OSPREY[39]). However, the functions of proteins are often driven by the interdependencies between residues (e.g., epistasis) in the protein[40,41], and not all the protein structures are solved. We thus explicitly modeled the pairwise interactions of all pairs of sites in a protein by extracting signals of evolutionary conservation from its homologous sequences or sequence families. We used a generative graphical model, fitted on the multiple sequence

alignment (MSA) of the homologous sequences, to uncover the underlying constraints or interdependencies that define the family of homologous sequences. These constraints are the results of the evolutionary process under natural selection and may reveal clues on which mutations are more tolerable or favorable than others. The generative model generates a sequence $\mathbf{x} = (x_1, ..., x_L)$ with probability $p(\mathbf{x}) = \exp[E(\mathbf{x})]/Z$, where $E(\mathbf{x})$ is the 'energy function' of sequence $\mathbf{x}$ in the generative model and $Z$ is a normalization constant. We applied CCMpred[34], which is based on a Markov random field (MRF) specification to model the residue dependencies in protein sequences. The energy function $E(\mathbf{x})$ of sequence $\mathbf{x}$ is defined as the sum of all pairwise coupling constraints $\mathbf{e}_{ij}$ and single-site constraints $\mathbf{e}_i$, where $i$ and $j$ are position indices along the protein sequence,

$$E(\mathbf{x}) = \sum_i \mathbf{e}_i(x_i) + \sum_{i \neq j} \mathbf{e}_{ij}(x_i, x_j) \qquad (1)$$

When the MRF model is fit to data with proper regularizations, the residue interactions in protein sequences are explained by the direct coupling terms $\mathbf{e}_{ij}$. It has been shown that the magnitudes of $\mathbf{e}_{ij}$ terms can accurately predict protein contacts[42] and 3D structures[43]. For a protein sequence with length $L$, we encoded its $i$-th amino acid $x_i$ by a vector, in which elements were set to the single-site term $\mathbf{e}_i(x_i)$ and pairwise coupling terms $\mathbf{e}_{ij}(x_i, x_j)$ for $j = 1, ..., L$ (Fig. 1c), and then dimensionality reduction techniques were used to project it into low rank ("Methods"). Encoding the protein sequence in this way directly incorporates the protein's evolutionary context, i.e., the effects of pairwise epistasis, which can inform machine learning models to predict the fitness of a sequence with single or higher-order combinatorial mutations.

In addition to the evolutionary sequence contexts specific to the protein of interest, global protein sequence contexts, i.e., those encoding structures and stabilities, can also inform our prediction model to predict the effects of mutations. For this purpose, we integrated general protein sequence representations from unsupervised protein contextual language models[13,17–19]. Using a large corpus of protein sequences such as UniProt and Pfam, a language model learns to predict the likelihood of a particular amino acid appearing at a position given all other amino acids surrounding it as context. During the training, the language model gradually changes its internal dynamics (encoded as hidden state vectors) to maximize the prediction accuracy. It was found that a wide range of protein-relevant scientific tasks, including secondary structure prediction, contact prediction, and remote homology detection, can be improved by using the hidden state vectors of a language model as input features to fine-tune a supervised model for the specific task[18,19]. Here, we also used the language model's hidden state vectors as another type of protein sequence representation for our prediction model to capture the global protein sequence context (Fig. 1c; "Methods"), which is a complement to our local evolutionary context representation.

The local and global evolutionary representations are jointly used to model the protein sequence of interest. A deep learning model (recurrent neural network) then takes these sequence representations as input and learns the sequence-to-function relationship. Quantitative functional measurements (e.g., fitness data measured by deep mutational scanning) are used to supervise the training of the deep learning model (Supplementary Fig. 1; "Methods").

**Accurate prediction of functional fitness landscape of proteins.** To validate the ECNet, we performed multiple benchmarking experiments to assess the ability of ECNet in predicting the functional fitness from protein sequences.

We first compared our evolutionary context representation to different representation schemes for protein sequences or mutations. Yang et al.[44] proposed to use a Doc2Vec model[45], pre-trained on ~500k UniProt sequences, to map an arbitrary sequence to a 64-dimensional real-valued vector. To directly test the utility of sequence representations, we used our deep learning model as the predictor for both our representation and the Doc2Vec representation of Yang et al. We compared the two approaches on the Envision dataset[35], composed of 12 DMS studies that generated fitness values of single amino acid variants of ten proteins ("Methods"). We found that ECNet consistently outperformed the approach of Yang et al on all the 12 datasets, with a relative improvement ranging from 16 to 60% in terms of the achieved Spearman correlation (Fig. 2a). Since the Doc2Vec representation was learned from the UniProt dataset, the information it captured is mostly general protein properties but not the dependencies in the sequence that determine functions. In

contrast, our evolutionary context representation explicitly models the epistasis of residue pairs in the sequence, which jointly influence the function in a non-independent way. This fine-grained information informed the prediction model to learn the sequence-function mapping more effectively and thus improved the prediction performance. We also compared our evolutionary context representation to the Envision model[35], which described a single amino acid substitution using 27 biological, structural, and physicochemical features. Compared to this approach, ECNet, without using these features, still improved the Spearman correlation for most of the proteins (Supplementary Fig. 2; Supplementary Table 1). As protein engineering focuses on identifying variants with improved properties than the wild type, we further evaluated the model performance using a classification metric (AUROC score), in which variants with higher function measurements than the wild-type sequence are defined as positive samples, and the remaining variants as negative samples. We observed similar improvements in AUROC scores for 11/12 protein DMS datasets (Fig. 2b; Supplementary Table 1). These results suggest that sequence contexts are more informative than the descriptors of mutated amino acids, which is critical in capturing the interdependencies between residues to predict the functions.

Next, we compared ECNet to other sequence modeling approaches for mutational effects prediction on a larger set of DMS datasets previously curated[15]. We first compared it to three unsupervised methods, including EVmutation[14], DeepSequence[15], and Autoregressive[46]. These methods trained generative models on homologous sequences and predicted the mutation effects by calculating the log-ratio of sequence probabilities of mutant and wild-type sequences. As expected, ECNet, predicting the mutation effects using a supervised predictor, outperformed these methods across almost all proteins (Fig. 3a), compared to EVmutation (median difference in Spearman correlation $\Delta\rho = 0.216$), DeepSequence (median $\Delta\rho = 0.196$), Autoregressive (median $\Delta\rho = 0.165$). There were only two proteins on which ECNet did not clearly outperform other unsupervised methods. This is likely due to the relatively small number of function measurements available that we can use to train the supervised predictor (1777 and 985 measurements, respectively; median: 2721 across all proteins). We expect that a more regularized prediction model will achieve improved prediction performance for proteins with a small set of function measurements. We also compared ECNet to two supervised methods. One is TAPE that uses the sequence representations learned by the protein language model[19] as input to train a neural network that has the identical model architecture as ECNet. The other is UniRep[17], which uses the output of its own language model to train a top model based on ridge regression. We found that ECNet, by combining global LM representations, local evolutionary representations, and the raw sequence as input, achieved higher correlations than TAPE (median $\Delta\rho = 0.089$) and UniRep (median $\Delta\rho = 0.109$) that used LM representations alone for nearly all proteins (Fig. 3b). We also performed an ablation analysis to dissect the performance of each representation component in our model's input and found that a model using joint representations outperformed a model using any individual representation (Supplementary Fig. 3). Furthermore, we simulated experiments where ECNet was trained on noisy training data and tested on noise-free data. We found that ECNet was robust against data noise (Supplementary Fig. 4). For example, ECNet's test correlation only decreased by 2% when the training data was perturbed by 10%. In contrast, a simple sequence representation such as one-hot encoding was impacted severely by data noise. Overall, tested on a large set of DMS data, ECNet significantly outperformed other sequence modeling methods,

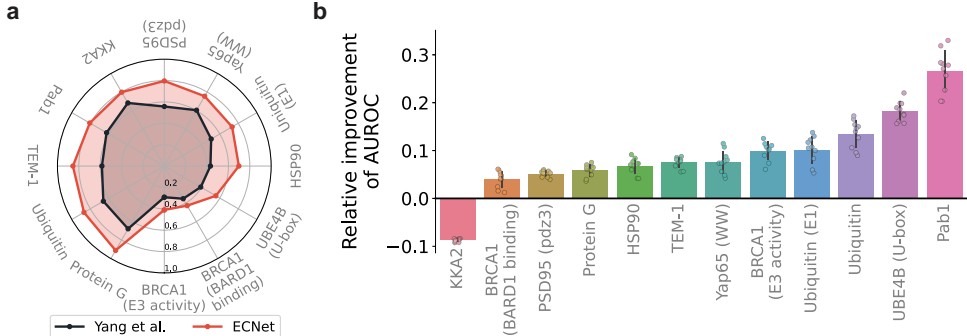

**Fig. 2 Comparisons to other protein variant representation methods. a** Comparison to the approach from Yang et al.[44] that represents protein sequences with fixed-length vector representations by training a Doc2Vec model on the UniProt database. Spearman correlation was used as the evaluation metric. Performances were evaluated using five-fold cross-validation. **b** Comparison to the Envision model[35] that represents a variant with 27 biological, structural, and physicochemical descriptors. AUROC (area under the receiver operating characteristics) was used as the evaluation metric to assess the ability of the model in identifying variants with improved function compared to the wild type. Relative improvements achieved by ECNet over the Envision model were shown in the bar plot. Performances were evaluated using ten trials of five-fold cross-validation. The bar plot represented the mean ± SD of the data.

either unsupervised or supervised (Fig. 3c; one-sided rank-sum test $P < 10^{-5}$), demonstrating its superior ability in predicting the fitness landscape of protein variants.

In addition to evaluating ECNet's performance of predicting fitness across all variants as shown above, we further designed an experiment to assess ECNet's ability to prioritize high-performing variants. To this end, we trained an ECNet model and applied it to predict and rank all variants in the randomly split test set based on their predicted fitness. We then calculated the fraction of the true top 100 variants that were ranked in the top $K$ predictions of ECNet. This experiment simulated the process in directed evolution where we want to identify and synthesize the most promising variants for screening, given a sequencing budget of $K$ variants[47]. On three DMS datasets of avGFP, GB1, and Pab1, we found that ECNet achieved higher recall (Supplementary Fig. 5a) and more efficiently discovered the variant with the highest fitness (Supplementary Fig. 5b) than UniRep and EVmutation. ECNet also achieved a 15–50× efficiency gain over a random sampling approach (Supplementary Fig. 5c), which is a widely used strategy in current directed evolution workflows. These results suggest that ECNet is an effective method to retrieve high-ranking variants for protein engineering and can potentially improve the efficiency of directed evolution in the laboratory.

As a supervised model, the performance of ECNet can be limited when the available DMS data is too scarce to train an accurate predictor. To address this challenge, we further built an unsupervised version of ECNet model that does not require any DMS data for training but is able to produce reasonably accurate predictions. Inspired by protein language models, we built unsupervised ECNet by training it on homologous sequences of the protein of interest using the language model objective. The predicted probability of an amino acid at a position was used as the proxy of fitness prediction ("Methods"). Tested on four DMS studies covering ten viral protein strains, unsupervised ECNet achieved an average Spearman correlation of 0.37 (Supplementary Fig. 6). This unsupervised variant of ECNet is particularly useful when the target protein is novel and has very few available DMS data. For example, we observed that unsupervised ECNet achieved reasonably good performance (mean Spearman correlation 0.36; Supplementary Fig. 7) on the DeepSequence dataset without using any DMS data as the supervised signal. In addition, using a small number of DMS data (e.g., 25% of available data of each protein) to train a supervised ECNet model substantially improved the prediction performance (mean Spearman correlation 0.54; Supplementary Fig. 7), and using the full DMS data further boosted the results (mean Spearman correlation 0.71;

Supplementary Fig. 7). We thus expect that unsupervised ECNet can select promising variants for screening in the first round of directed evolution, after which the screening data can be used to train a supervised ECNet for later rounds to improve the model accuracy and prioritize improved variants.

**Generalization to higher-order variants from low-order variants data.** Construction and screening of higher-order variants can require a significant amount of experimental effort and time. As a result, fitness measurements of single mutants were more prevalent in existing DMS studies as compared to those of double or higher-order mutants. It is thus highly desired in protein engineering that a machine learning model trained on fitness data of low-order variants can also accurately predict the fitness of higher-order variants. As such, the model can fully leverage the fitness data of screened low-order variants and prioritize higher-order variants that are likely to exhibit improved properties for the next round of directed evolution.

We thus assessed ECNet's performance on predicting the fitness of higher-order variants when only lower-order data were used for model training. We collected the fitness measurements of both single and double mutants of six proteins from previous DMS studies[33,48–52]. We then trained our prediction model using single mutant data only and tested its performance on double mutants. ECNet achieved Spearman correlations ranging from 0.73 to 0.94 for the six proteins and outperformed the TAPE and the EVmutation methods (Fig. 4a), suggesting its generalizability to the prediction of higher-order variants from low-order variants data. We also observed that the increased diversity of fitness landscape in the training data improved the prediction performance. For example, to predict the fitness of quadruple mutants of the avGFP protein[53], we trained separate models using the fitness data of single, double, triple, or all three orders of mutants. The test results suggested that a model trained on higher-order mutation data (from single to triple) achieved an increasing prediction performance, and the union of all-order mutation data further improved the prediction (Fig. 4b). To further assess ECNet's ability, we used orthogonal data containing sequences of 146 TEM-1 variants that are known to be inhibitor-resistant ("Methods"). Sequences in this data contain two to ten (mean 3.3) amino acid substitutions compared to the TEM-1 protein. Based on these sequences, we generated ten times more random variants by enumerating all mutation combinations restricted to the positions where mutations were introduced in the 146 variants ("Methods"). We then trained our model on fitness data of TEM-

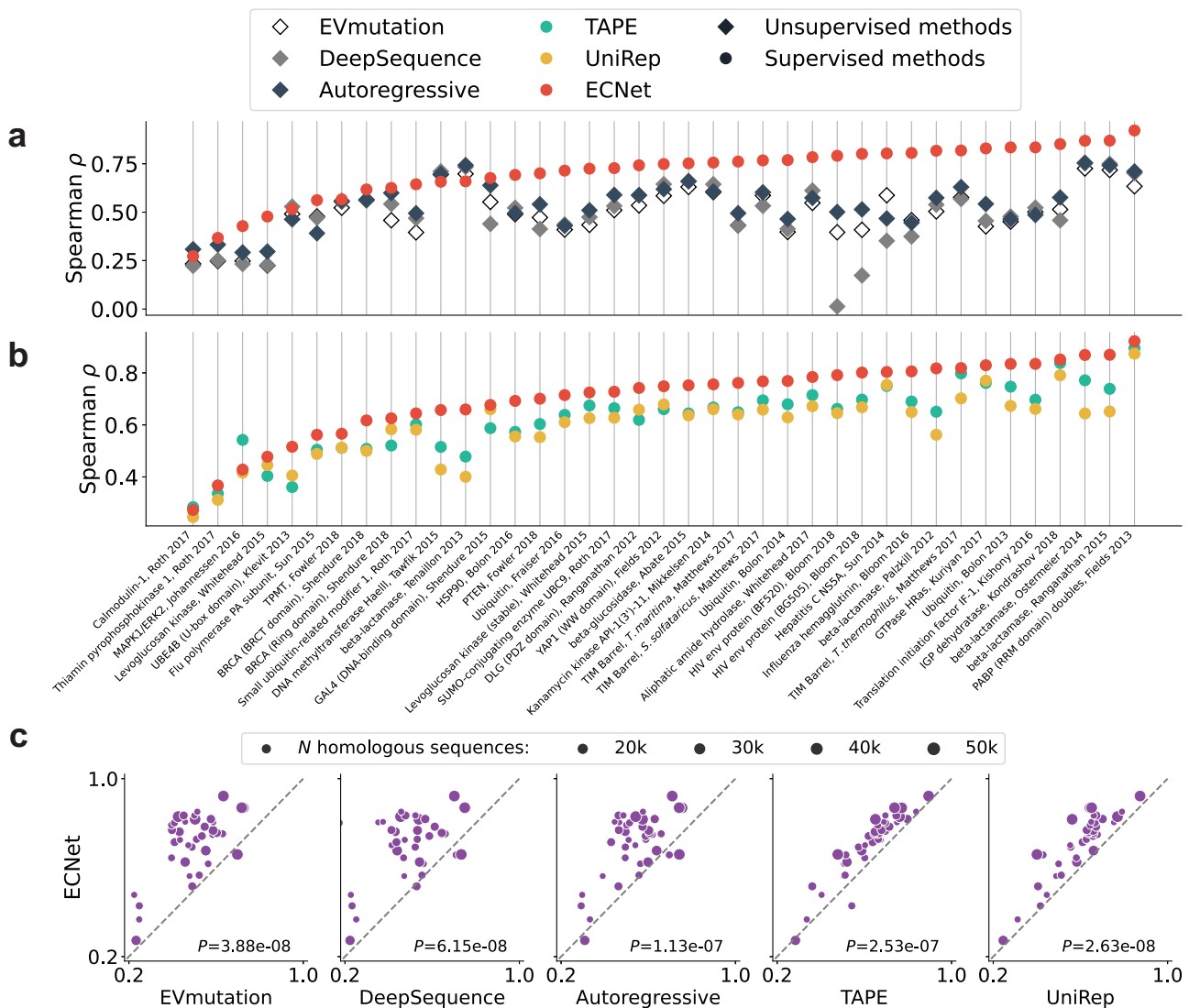

**Fig. 3 Comparisons to other sequence modeling approaches for mutation effects prediction. a** Comparisons to three unsupervised generative models, EVmutation, DeepSequence, and Autoregressive. **b** Comparisons to two supervised models (UniRep and TAPE) that use a pre-trained protein language model to learn protein sequence representations and fine-tunes a supervised predictor using functional measurements. **c** Pairwise comparisons between ECNet and other methods. Each data point represents the performance on the DMS data of a protein and the dot size is proportional to the number of homologous sequences of the protein. Spearman correlation was used as the evaluation metric for all results in this figure. One-sided rank-sum test was used to test the statistical significance.

1 single mutants data and used it to predict the fitness of the 146 TEM-1 variants as well as the randomly generated variants. We found that ECNet distinguished the inhibitor-resistant variants from the random variant background (Fig. 4c; mean predicted fitness 0.79 vs. 0.48; one-sided rank-sum test $P < 10^{-5}$). This orthogonal validation further demonstrates the generalizability of ECNet, even trained on single mutants data, to the prediction for higher-order mutants.

It was shown that mutations within the sequence can have non-independent effects (epistasis) on fitness[40,54]. The double mutant fitness $f_{ij}$ may not always be equal to the sum of constituent single mutant fitness $f_i + f_j$, where $f$'s are the (log-transformed) experimentally measured fitness of variants. Epistasis ($\epsilon$) is quantified as the difference between the experimentally measured fitness and the expected fitness: $\epsilon = f_{ij} - (f_i + f_j)$. To analyze whether ECNet captures the interdependencies between mutations, we correlated the observed epistasis $\epsilon$ with predicted epistasis $\hat{\epsilon}$, which is defined as

$\hat{\epsilon} = \hat{f}_{ij} - (\hat{f}_i + \hat{f}_j)$ where $\hat{f}$'s are predicted fitness. Compared with EVmutation that explicitly models epistasis using a generative model, the epistasis predicted by ECNet better correlated with the observed epistasis (Fig. 4d; one-sided rank-sum test $P < 10^{-5}$). The epistasis captured by ECNet was also more accurate or comparable to that of TAPE (Fig. 4d). These results suggest that ECNet captured the residue dependencies within sequences more accurately, and thus resulted in the superior prediction performances reported above.

**Engineering of TEM-1 β-lactamase using ECNet.** To experimentally validate its utility in protein engineering, we applied ECNet to prioritize new higher-order TEM-1 β-lactamase variants that are likely to have improved fitness compared to the wild type. We trained ECNet using DMS data reported in previous studies[51,55]. The datasets curated the fitness measurements of nearly all point-mutation variants and 12% of possible consecutive double-mutation variants of TEM-1. We performed in

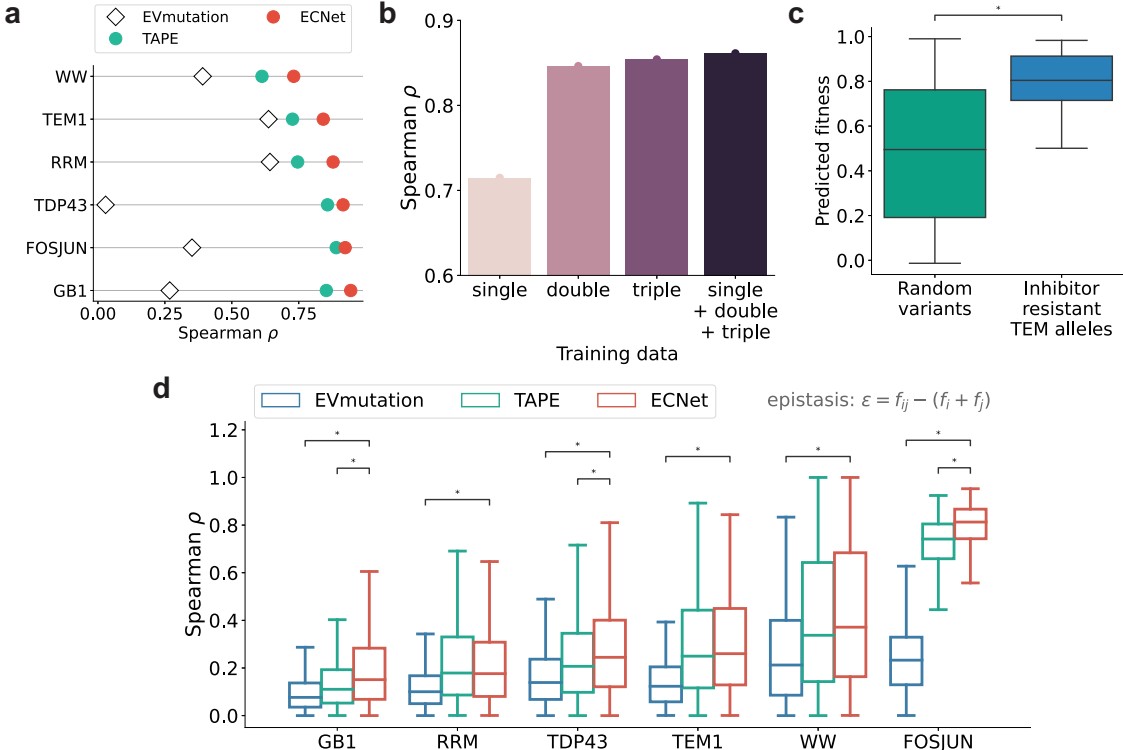

**Fig. 4 Accurate prediction of higher-order variants using a model trained on lower-order variants. a** Prediction of the fitness of double mutants. For supervised methods (ECNet and TAPE), the prediction models were trained using fitness measurements of single mutants. **b** Prediction of quadruple mutants of avGFP using models trained on single, double, triple, and all three types of mutants. **c** The predicted fitness values of inhibitor-resistant TEM-1 variants ($n = 146$) were significantly higher (one-sided rank-sum test $P = 5.1 \times 10^{-31}$) than those of randomly generated background variants ($n = 1460$). **d** Spearman correlation of experimentally measured epistasis and predicted epistasis for double-mutation variants of GB1 ($n = 4455$), RRM ($n = 2700$), TDP43 ($n = 5166$), TEM-1 ($n = 841$), WW ($n = 1680$), and FOSJUN ($n = 3072$). In comparison to EVmutation, the Spearman correlations achieved by ECNet were significantly higher for all six proteins (one-sided rank-sum test $P$ values: GB1: $6.7 \times 10^{-72}$, RRM: $4.7 \times 10^{-35}$, TDP43: $4.0 \times 10^{-7}$, TEM-1: $1.8 \times 10^{-18}$, WW: $6.7 \times 10^{-17}$, FOSJUN: $1.0 \times 10^{-80}$). In comparison to TAPE, ECNet was comparable for proteins RRM, TEM-1, and WW and achieved significantly higher correlations for proteins GB1, TDP43, and FOSJUN (one-sided rank-sum test $P$ values $2.4 \times 10^{-19}$, $4.2 \times 10^{-9}$, and $2.2 \times 10^{-51}$, respectively). In box plots, the midline represents the median, the lower and upper hinges of the boxes correspond to the 25th and 75th percentiles, and the whiskers extend to 1.5 times the interquartile range from the hinges. The asterisk symbol * indicates $P$ values $<10^{-5}$.

silico mutagenesis for several function-related sites of TEM-1 curated in the literature and their higher-order recombinations ("Methods"). We then applied ECNet to predict the fitness for all variants generated from the in silico mutagenesis. After removing structurally unstable variants, we selected 37 variants that were ranked at the top by either the standard ECNet model or an ensemble version of ECNet, which averages predictions of multiple replicates of ECNet models ("Methods"). The 37 top-performers were novel TEM-1 variants and did not overlap with any variants in our training data or functional TEM-1 variants we collected from the literature. Despite that the training data only covered single and consecutive double mutants, these 37 variants sampled a diverse combination of mutation sites and contained higher-order mutants ranging from 2 to 6 mutations (Supplementary Data 1).

We created those 37 variants and nine previously reported TEM-1 mutants which had demonstrated strong resistance against ampicillin to serve as positive controls[51,55] (Supplementary Data 2). We plated the library containing these 37 variants and positive controls on LB agar plates with ampicillin of various concentrations (300, 1500, and 3000 μg/mL) to test their resistance capacity against ampicillin. Further, PacBio sequencing was performed to determine the relative abundance of these variants before and after selection, as a proxy of their fitness (Fig. 5a; "Methods"). The fitness of each mutant at a certain ampicillin concentration was calculated based on the ratio of the

relative abundance of the mutant to wild-type TEM-1 in the plate with the related concentration of ampicillin and the relative abundance of the mutant to wild-type TEM-1 in the plate without ampicillin ("Methods" and Supplementary Data 1). We observed that most of the variants prioritized by ECNet demonstrated improved fitness as compared to the wild type (Fig. 5b). The improvements were observed at various concentrations of ampicillin (300, 1500, and 3000 μg/mL) and were reproducible across different replicates. Notably, ECNet has identified variants that improved the wild-type fitness by up to ~8-fold, which was substantially higher than the best performers we had in the training data (positive controls in Fig. 5b). We also found that the ensemble model of ECNet achieved robust predictions, with a mean hit rate (fraction of predicted variants with fitness higher than the wild type) 0.52, 0.91, and 0.94 for concentrations 300, 1500, and 3000 μg/mL, respectively (Fig. 5c).

Despite being trained on the data of single mutants and consecutive double mutants, ECNet prioritized novel and higher-order TEM-1 mutants that showed improved resistance against ampicillin. The validation results suggested that the evolutionary contexts enable ECNet to discover higher-order mutants that have not been observed in the training data. The results also demonstrate the potential of ECNet to be integrated into the existing protein engineering workflows to guide the discovery of enhanced variants.

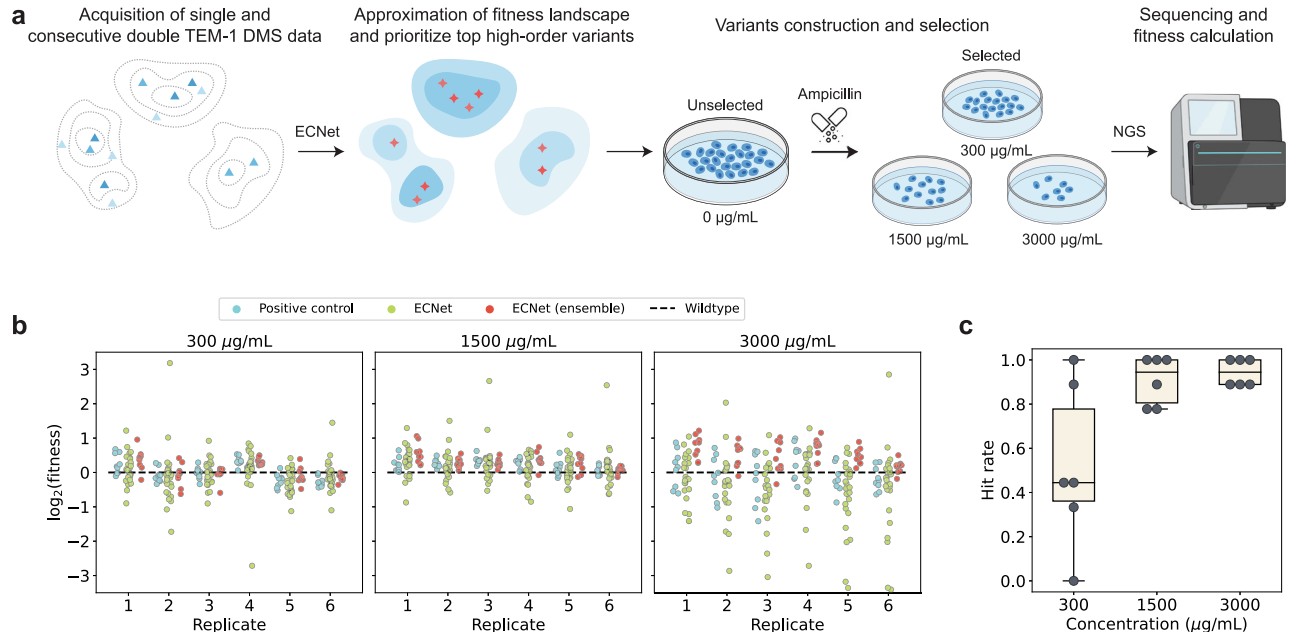

**Fig. 5 ECNet enables the rapid engineering of TEM-1. a** The workflow of using ECNet to predict enhanced TEM-1 variants. The ECNet model was trained on fitness data of single and consecutive double TEM-1 mutants and applied to prioritized higher-order mutants; top-ranked TEM-1 variants were constructed and their fitness (resistance against ampicillin) was measured (DMS: deep mutational scanning; NGS: next-generation sequencing). **b** Fitness values of predicted TEM-1 variants at different ampicillin concentrations. Results from six replicates are shown. Fitness values of variants prioritized by an ensemble version of ECNet (averaged predictions of multiple replicates of ECNet models) are colored separately. Top-performing single or consecutive double mutants in the training data are labeled as positive controls. The Black dashed line represents the fitness of wild-type TEM-1. **c** Hit rate (fraction of predicted variants with fitness higher than the wild type) of the ensemble ECNet model. Each point represents a replica ($n = 6$ replicates in total). The midline of box plots represents the median, the lower and upper hinges of the boxes correspond to the 25th and 75th percentiles, and the whiskers extend to 1.5 times the interquartile range from the hinges.

## Discussion

A critical challenge in machine learning-guided protein engineering is the development of a machine learning model that accurately maps protein sequences to functions for unseen variants. While models have been developed for the qualitative classification of protein sequences into function classes, such as those in the Critical Assessment of Functional Annotation (CAFA) challenge[10], in protein engineering prediction models are required to provide a more fine-grained characterization of protein functions, which distinguishes the quantitative function levels of closely related sequences (e.g., single-site mutants of wild-type protein with sequence similarity >99%). The function prediction in protein engineering is also different from predicting the deleteriousness[56] or instability[38] of variants—to assist protein engineering, the machine learning model needs to prioritize variants that are not only structurally stable and non-deleterious but also with enhanced properties. Furthermore, as the protein sequence space is tremendous in size, it is desired to have a machine learning model that navigates the fitness landscape effectively and can generalize from regions of low-order variants to regions in the landscape where higher-order variants with improved function may exist. All these factors render it uniquely challenging to develop a machine learning model that can be used to guide protein engineering strategies such as directed evolution and rational design.

In this work, we have presented a high-performance method, ECNet, that predicts protein function levels from sequence to facilitate the process of protein engineering. Supervised machine learning models have been explored recently to predict protein sequence-function relationships[47,57–59]. As in those studies, in this work, we mainly focused on improving the protein function by introducing point mutations, while introducing insertion/

deletion was also explored in other work[60]. Our machine learning model uniquely used a biologically-motivated sequence modeling approach to learn the sequence-function relationship, leading to superior performances in predicting the fitness of protein variants. Benchmarked on a large set of deep mutational scanning studies, ECNet outperformed multiple existing machine learning models for protein engineering. Further, ECNet accurately captures the epistasis effects of mutations within protein sequences and can be generalized to predict higher-order mutants' functions by learning from the data of lower orders. We applied ECNet to engineer TEM-1 β-lactamase and experimentally validated that it successfully identified variants with enhanced ampicillin resistance with high hit rates.

ECNet's prediction performance is impacted by the MSA characteristics and DMS data properties. For example, ECNet predicts better for proteins with more homologous sequences (Supplementary Figs. 8 and 9a), for sites that are more conserved within a protein (Supplementary Fig. 9b), and for proteins that have a more complete DMS dataset (Supplementary Fig. 9c-d). In our additional tests that used both a sequence site-wise strategy and a per-site AA-wise train/test split strategy[61] to assess ECNet's generalizability, we found that, despite the challenging setting, ECNet still outperformed the DeepSequence when predicting for new mutation sites (Supplementary Fig. 10). Further investigation revealed that the exploration–exploitation trade-off of training data also influenced the model performance (Supplementary Fig. 11). This implies that the design of more effective training data should be taken into account when developing ML algorithms to assist protein engineering, especially when the experimental test budget is limited[62].

We expect ECNet to be a practical tool for ML-guided protein engineering. In a round of directed evolution, the sequence-to-

function model can be applied, potentially coupled with other sequence design algorithms[63–65], to select the next set of variants to screen. In addition, given its generalizability to higher-order mutants from lower-order mutants, the model can fully leverage the screening data of low-order mutants, including that of both improved and unimproved variants, generalize to distant regions in the fitness landscape where higher-order variants with improved properties may exist, and prioritize promising higher-order mutants to screen in the next round, in which the screened data can be used to further improve the model, hereby forming an iterative loop of directed evolution to discover improved variants.

## Methods

**Datasets**. We collected multiple large-scale deep mutational scanning (DMS) datasets and random mutagenesis datasets curated by previous publications.

*Envision dataset.* We first collected 12 DMS studies from Gray et al.[35], covering ten proteins and 28,545 fitness measurements of single amino acid variants. The fitness values were normalized such that wild-type-like variants having scores of one, and variants that are more (less) active than the wild type having scores greater (less) than one.

*DeepSequence dataset.* We also collected a set of DMS datasets compiled by Riesselman et al.[15]. We excluded a study of RNAs since it is out of the scope of this study. The resulting set consists of 39 DMS studies across 33 proteins. Most of these studies (37/39) provide the function values of single amino acid variants, and two studies provide the functional measurements of higher-order mutants. The functions measured in these studies include growth rate, enzyme function, protein stability, and peptide binding.

*Single and double mutants datasets.* To test the ability of ECNet to predict epistasis, we compiled multiple DMS studies that contain the fitness values of both single and double amino acid variants. We obtained the DMS data of the GB1 domain, WW domain, RRM domain, and FOS–JUN heterodimer from Rollins et al.[30], and the prion-like domain of TDP-43 from Bolognesi et al.[52]. A set of fitness of TEM-1 consecutive double mutants was also obtained from Gonzalez et al.[51]

*Higher-order avGFP mutants dataset.* We also collected a higher-order mutant dataset[53] to assess ECNet's generalizability to predict the effect of even higher-order variants. This study[53] systematically assayed the local fitness landscape of the green fluorescent protein from *Aequorea victoria* (avGFP) by measuring the fluorescence of ~50k derivatives of avGFP, with each sequence containing 1–15 amino acid substitution mutations.

*Inhibitor-resistant TEM-1 variants.* We compiled a list of TEM-1 variants that have been found to be inhibitor-resistant with supporting evidence in previous studies. The list was downloaded from https://externalwebapps.lahey.org/studies/TEMTable.aspx (see "Data availability" for accession). We excluded variants for which mutation information was labeled as "Not yet released". This resulted in 146 sequences that mostly contained two to five and up to ten mutations (average 3.3 mutations per sequence). To generate a list of random candidate variants for enhanced TEM-1 variants prioritization, we enumerated all combinations of amino acid mutations on all or a subset of the positions where mutations were introduced in this 146-sequence list. In total, we obtained 18,937 randomly generated candidate variants for TEM-1 variants prioritization.

*Homologous sequences and fitness data of viral proteins.* We used the homologous sequences of each viral protein collected in Hie et al.[25] as the training data of the unsupervised ECNet, including 44,851 unique influenza A hemagglutinin (HA) amino acid sequences observed in animal hosts, 57,730 unique HIV-1 Env protein sequences, and 4172 unique Spike and homologous protein sequences. We used the fitness data collected in Hie et al.[25] to validate the unsupervised ECNet. The fitness data includes replication fitness of HA H1 WSN33 mutants from Doud and Bloom[66], replication fitness of six HA H3 strains (Bei89, Bk78, Bris07, HK68, Mos99, and NDako16) from Wu et al.[67], replication fitness of HIV Env BF520 and BG505 mutants from Haddox et al.[68], and $K_d$ binding affinities between SARS-CoV-2 mutants and ACE2 from Starr et al.[69].

**Inference of evolutionary couplings from multiple sequence alignments**. We first searched homologous protein sequences of a given protein using HHblits available in the hh-suite[70]. We used the wild-type sequence of the given protein as the query sequence and searched against the uniclust-30 database (version uniclust30_2018_08) for three iterations. We used a maximum pairwise sequence identity of 99% and a coverage cutoff of 50%. Other parameters were set as default. The search results were formatted to the A3M multiple sequence alignment (MSA) format.

To identify the co-evolutionary residue pairs in a protein, we used a statistical model to exploit the evolutionary sequence conservation and model all pairwise interdependencies of residues. The model identifies the evolutionary couplings by learning a generative model of the MSA of homologous sequences using a Markov random field. Given the MSA of homologous sequences, the couplings are learned by maximizing the likelihood of observed sequences in the MSA, which is defined as

$$L(\mathbf{e}) = \frac{1}{Z} \prod_{n=1}^{N} \prod_{i=1}^{L} \left[ \exp\left( \mathbf{e}_i(x_i^n) + \sum_{j=1, j \neq i}^{L} \mathbf{e}_{ij}(x_i^n, x_j^n) \right) \right] \quad (2)$$

where the single-site constraints $\mathbf{e}_i$ and the pairwise coupling constraints $\mathbf{e}_{ij}$ are parameters of the model, $x_i^n$ is the $i$-th amino acid in the $n$-th sequence, $Z$ is the normalization constant, $N$ is the number of homologous sequences and $L$ is the number of columns in the MSA (number of amino acids in the query sequence). The direct optimization of this likelihood is computationally intractable due to the computation of the normalization constant that increases exponentially—$20^L$ sequences need to be considered. It was thus adopted to maximize the site-factored pseudo-likelihood of the MSA, which has a running time complexity $O(NL^2)$ where $N$ is the number of sequences in the MSA. We refer the interested readers to previous studies[34,71,72] for the details of the optimization. In this work, we used CCMPred[34], a GPU-based algorithm maximizing the pseudo-likelihood (plus regularization terms), to optimize the generative model. The evolutionary couplings are learned as parameters of the Markov random field.

**Local evolutionary context representation with evolutionary couplings**. By fitting the graphical model to the MSA of homologous sequences of a protein, we obtained the coupling matrix $\mathbf{e}_{ij}$ that quantifies the co-constraints of all possible $20^2$ amino acid combinations between positions $i$ and $j$ in the sequence. In particular, the term $\mathbf{e}_{ij}(x_i, x_j)$ is the pairwise emission potential of the Markov random field for amino acid $x_i$ occurring at position $i$ while amino acid $x_j$ occurring at position $j$. We used the site preference vector $\mathbf{e}_i$ and the coupling matrix $\mathbf{e}_{ij}$ to construct a data representation that encodes the co-evolution information of a protein.

Specifically, the $i$-th amino acid $x_i$ in the protein was represented by an $(L+1)$-long 'local evolutionary representation':

$$\mathbf{v}_i = [\mathbf{e}_i(x_i), \mathbf{e}_{i1}(x_i, x_1), \mathbf{e}_{i2}(x_i, x_2), ..., \mathbf{e}_{iL}(x_i, x_L)]. \quad (3)$$

The full representation of a protein sequence was thus obtained by stacking local evolutionary representations for all positions, resulting in an $L$ by $(L+1)$ matrix. As we have shown, the pairwise potentials in the matrix $\mathbf{e}_{ij}$ correlated with the fitness measured in DMS experiments (Fig. 1a, b). We thus expect that using the local evolutionary representations derived from $\mathbf{e}_i$ and $\mathbf{e}_{ij}$ as a data representation of amino acids will inform the sequence-to-function prediction model to better capture the residue dependencies and the sequence-to-function relationship.

The length of the local evolutionary representation is roughly equal to the length of the protein sequence, which may raise an overfitting issue when the protein length is long while the number of functional measurements used as training data is low. Therefore, we used a dimensionality reduction approach to transform the $(L+1)$-long vector into a fixed-length $d$-dimensional vector ($d < L$), where $d$ is independent of the length of the protein sequence. This is done by applying a linear layer in the neural network to reduce the dimensionality of local evolutionary representations $\mathbf{v}_i$. Hereinafter, we will refer to the transformed vector $\mathbf{v}_i$ as local evolutionary representation unless otherwise specified.

**Pre-trained protein sequence representation model**. Very recently, self-supervised models have provided powerful protein sequence representations that facilitate scientific advances, including protein engineering, structure prediction, and remote homology detection. These language models[13,17–19], without using labeled data, are trained on natural sequences from large protein databases such as Pfam[28] and UniProt[27] to predict the next amino acid character given all previous amino acid characters in the protein sequence or predict randomly masked amino acids using the rest as given context. During the model training, these models progressively adapt their parameters to maximize the prediction accuracy, resulting in a representation of protein sequences that capture intrinsic semantics in protein sequences and interdependencies among amino acids.

In this work, we integrated the amino acid representations produced by a transformer model in TAPE, one of the most powerful self-supervised sequence representation models[19]. The representations capture the global evolutionary context from the massive protein sequence data the model was trained on, which is complementary to our evolutionary representations that capture the local evolutionary context specific to the target protein. The TAPE model applied a Transformer architecture[73] and was trained on Pfam data to predict a masked amino acid using the remaining ones as input. We downloaded the pre-trained weights of the TAPE model from https://github.com/songlab-cal/tape. For an input sequence, TAPE generates a 768-dimensional vector representation for each amino acid. We refer to the reprojected TAPE representations as global evolutionary representations.

## Sequence-to-function neural network model

*Model architecture.* We built a deep learning model for the sequence-to-function prediction. The model receives as input features (amino acid characters and evolutionary representations) of the protein sequences and produces the predicted functional measurements of proteins as output. The backbone of our model is a bidirectional long short-term memory network (BiLSTM)[74] integrated with a two-layer fully-connected neural network. Hyperparameters of the model were decided through a grid search in an independent experiment (see "Training details"). Amino acids in the input sequences were one-hot encoded and passed through a 20-dimensional embedding layer. The amino acid embeddings were then concatenated with the evolutionary representations position-wisely before being input to the LSTM module. We used a single-layer LSTM with a hidden dimension $d_L$ as the default setting in this work. One hidden state vector was produced by the LSTM for every amino acid in the sequence. To integrate the TAPE representations into our model, we reprojected them to $d_p$-dimensional vectors using a linear fully-connected layer, which were then concatenated with the hidden state vector produced by the LSTM model for each amino acid. We summarized these concatenated vectors into a single vector using a weighted averaging approach, where the averaging weights were learned from the data by using a self-attention layer[73]. This vector was then passed to a top module to predict the functional measurements. The top module is a two-layer fully-connected neural network with tanh activation. The hidden dimensions of the two layers were set to $d_h$ and 1, respectively. To facilitate the model training, we added a batch normalization layer[75] before the fully-connected layers. We also applied a dropout[76] layer after the first fully-connected layer to prevent overfitting. To improve the model's robustness and prediction accuracy, we used an ensemble approach to output the prediction, in which three replicas of ECNet models were trained using the same hyperparameters and training data, and their output scores were averaged as the final prediction.

*Training details.* We cast the task of predicting the functional values of proteins as a regression problem, and the objective was to minimize the difference between the predicted and experimentally measured functional values. We trained our deep learning model using the Adam optimizer[77] with default parameters. Mean squared error (squared $L_2$ norm) was used as the loss function. To select the hyperparameters of ECNet, we performed a small-scale grid search using the training data of a protein, such that 7/8 of the training data was used to train a model with a specific set of hyperparameters, and the remaining 1/8 data was used as the validation set to select the hyperparameters. The test set was not used for hyperparameter selection. We tested the LSTM's dimension of $d_L$ = 32, 64, and 128, the top layer dimension of $d_h$ = 32, 64, and 128, the reprojected embedding dimension of $d_p$ = 128 and 256. In general, we found that $d_L$ = 128, $d_h$ = 128, and $d_p$ = 128 are reasonably good defaults and can be used for a new protein. Nevertheless, a careful grid search of hyperparameters for the new protein would further improve the model performance. Unless otherwise specified, the batch size was set to 128 and the maximum number of training epochs was set to 2000 with an early stop if the performance has not been improved for 1000 epochs. Model training was performed on an Nvidia TITAN X GPU. The time required to train a single model depends on the training data size of each protein, ranging from 0.5 to 6 h. For the ensemble model with three replicas, the required time thus ranges from 2 to 20 h.

*Auxiliary classification objective.* While the prediction of functional measurements is a regression problem by definition, the skewed distribution of the training data may lead to a biased predictor. For example, in the Envision dataset[35], only 18% of TEM-1 variants are more active than the wild-type sequence (positive effects) while the remaining are less active than the wild-type sequence (negative effects). In this case, a model optimized using a regression objective (e.g., minimizing the mean squared error) tends to fit the negative effects more but be less sensitive to the error from the prediction of positive effects. However, the main goal of machine learning-guided protein engineering is to identify the variants with an enhanced property than the wild-type sequence. Hence, it is critical to mitigating this type of bias in the prediction model. We addressed this issue by introducing an auxiliary classification objective. We binned the functional measurements using their 10-quantiles as breakpoints, i.e., grouping the measurements into 10 bins with equal size. In the model training, we encouraged the model to accurately predict not only the absolute functional measurement but also which bin the measurement is in. Jointly, the classification objective forces the model to treat each bin of functional measurements equally and the regression objective forces the model to predict the measurements as close to the observed values as possible. In the implementation, we added a second top module into the deep learning model, which also receives the summarized LSTM hidden state vector as input and its output is ten numbers indicating the predicted probability that the measurement should fall in each of the bins. The overall loss function is $L = L_r + \alpha L_c$ where $L_r$ is the loss of the regression objective, $L_c$ is the cross-entropy loss of a ten-class classification, and $\alpha$ is a constant used to balance the scales of the two losses, which was set as $\alpha = 0.1$ in this work. We used this hybrid loss when training the model for prioritizing novel TEM-1 variants and used the regression loss for other benchmarking experiments.

## Unsupervised ECNet based on language model training

While the vanilla ECNet is a supervised model and requires function or fitness data to train the predictor, we also developed an unsupervised extension of ECNet that does not need any direct fitness measurements as training data but is still able to produce reasonably accurate predictions. This unsupervised model is useful when the fitness data of a protein is unavailable or not sufficient to train an accurate supervised predictor. The predictions of the unsupervised ECNet can be used as an approximation of fitness and guide the selection of variants to screen in the first round of directed evolution, after which the experimental screening data can be used to train a more accurate, supervised ECNet model.

The main idea of the unsupervised ECNet is to train a model that learns the evolutionary preferences from the homologous sequences of the protein of interest. Those homologous sequences are the results of long-course evolution and might reveal evolutionary preferences about which mutations are more viable or tolerable than others. This approach is motivated by the recent advances of deep learning for human languages, in which algorithms called language models are developed to learn intrinsic semantics and grammar constraints of natural languages like English from large text corpora.

The model architecture of the unsupervised ECNet is also based on a bidirectional LSTM (BiLSTM), as in the supervised ECNet, but with a different training objective. Here, we use an objective similar to that used in Hie et al.[25] to train a protein language model. Precisely, we are given a protein sequence $\mathbf{x} = (x_1, ..., x_L), x_i \in X, i \in [L]$, where $X$ is the alphabet of all possible amino acids. Let $\tilde{x}_i$ denote a point-mutation at position $i$ and the mutated sequences $x(\tilde{x}_i) = (x_1, ..., x_{i-1}, \tilde{x}_i, x_{i+1}, ..., x_L)$. The language model aims to predict the probability of an amino acid appearing at a position considering its surrounding context, i.e., $p(x_i | x_{[L]\setminus\{i\}})$, where $x_{[L]\setminus\{i\}} = (x_1, ..., x_{i-1}, x_{i+1}, ..., x_L)$ represents the sequence context. The context is encoded using a latent real-valued vector $\mathbf{z}_i = f_e(x_{[L]\setminus\{i\}})$, where $f_e : X^{L-1} \to \mathbb{R}^D$ is an embedding function that maps discrete sequences into a $D$-dimensional continuous space. Here the embedding function was instantiated by a bidirectional LSTM neural network and the outputs of the final LSTM layers were concatenated to form the embedding vector, i.e.,

$$\mathbf{z}_i = [\text{LSTM}_f(g_f(x_1, ..., x_{i-1})); \text{LSTM}_r(g_r(x_{i+1}, ..., x_L))] \quad (4)$$

where $g_f$ is the output of preceding layers that proceed the input in the forward direction, $\text{LSTM}_f$ is the final layer of the forward-directed LSTM, and $g_r$ and $\text{LSTM}_r$ are defined similarly but for the reverse direction. The embedding vector $\mathbf{z}_i$ is transformed into a probability through a learner transformation and a softmax function, i.e.,

$$p(x_i | x_{[L]\setminus\{i\}}) = p(x_i | \mathbf{z}_i) = \text{softmax}(\mathbf{W}\mathbf{z}_i + \mathbf{b}) \quad (5)$$

where $\mathbf{W}$ and $\mathbf{b}$ are learned parameters. We used a two-layer BiLSTM with 256 units in this work.

We demonstrated the utility of the unsupervised ECNet on viral proteins. We trained three unsupervised ECNet models for influenza HA, HIV Env, and SARS-CoV-2 Spike proteins using the unsupervised ECNet model. One epoch in the training consisted of the prediction of every token at all positions and all sequences in the training set. The output probability $p(x_i | x_{[L]\setminus\{i\}})$ is used as the predicted score and to correlate with the fitness score of a mutant. If a variant has multiple mutations, the product of the probabilities of the individual point mutations was used as the predicted score of unsupervised ECNet.

## Baseline methods

We compared ECNet against several existing baseline methods, including supervised and unsupervised models.

*Yang et al. (Doc2Vec).* Yang et al.[44] proposed a learned protein embedding to represent a protein sequence in a 64-dimensional vector using a Doc2Vec model[45] trained on the UniProt database. The representation vector is used as the input feature to fit a Gaussian process-based regressor to predict the functional measurement. Following a previous work[17], we also used the four best-performing models as chosen in Yang et al.[44], including the original model (k = 3, w = 7), the scrambled model (k = 3, w = 5), the random model (k = 3, w = 7), and the uniform model (k = 4, w = 1). The pre-trained models were downloaded from http://cheme.caltech.edu/kkyang/models/ and protein representation vectors were generated using the code available at https://github.com/fhalab/embeddings_reproduction. The best performance across the four models was reported as the final performance of the Doc2Vec model.

*Envision.* Envision is a supervised method proposed in Gray et al.[35] that predicts the functional measurements of protein variants. Each variant was annotated with 27 biological, structural, and physicochemical features, which were used as input to train a gradient boosting regression model using large-scale mutagenesis data. We downloaded the source code of Envision from https://github.com/FowlerLab/Envision2017.

*EVmutation.* EVmutation is an unsupervised statistical model proposed by Hopf et al.[14]. It explicitly models the co-variations between all pairs of residues in the protein by fitting a pairwise undirected graphical model to the multiple sequence alignment (MSA) of all homologous sequences of the protein of interest. The model then quantifies the effect of single or high-order substitution mutations

using the log-ratio of sequence probabilities between the mutant and wild-type sequences. In this work, we used the workflow implemented in EVcouplings (https://github.com/debbiemarkslab/EVcouplings) to generate the predictions of EVmutation.

*DeepSequence*. Similar to EVmutation, DeepSequence[15] is also a generative model that predicts the effects of mutations in an unsupervised manner. However, unlike EVmutation explicitly modeling pairwise dependencies, DeepSequence uses a latent model, fitted on the MSA of homologous sequences of a protein, to capture higher-order dependencies of residues in the protein. The effects of mutations are also predicted by the log-ratio of mutant likelihood to wild-type likelihood.

*Autoregressive model*. Generative models of protein sequences such as EVmutation and DeepSequence are dependent on the alignment of homologous sequences, which may introduce artifacts and lose important information caused by indels in the alignment. A generative autoregressive model was proposed by Riesselman et al.[60] to predict the mutation effects in protein sequence, without the requirement of multiple sequence alignment.

*TAPE*. We used TAPE[19], a language model (LM) trained on Pfam sequences to generate global context representations of protein sequences. We extracted the hidden state vectors, one for each amino acid in the sequence, from the TAPE model. We used the same top module as in our model (i.e., self-attention layer and fully-connected layers) to take the representations as input and predict the functional measurements.

*UniRep*. UniRep[17] first trains an unsupervised protein language model on UniRef50 sequences. The model is then fine-tuned using homologous sequences of a studied protein (called evotuning). The model is used to generate a vector representation for each protein sequence. These representations are used as the input of a top supervised model such as ridge or LASSO regression to predict the fitness of mutants.

*CSCS*. CSCS[25] is an unsupervised model that is specifically designed to predict viral escapes. It also trains a language model on viral protein sequences and computes two scores to quantify the effect of a mutation, one is the grammaticality of the mutation, defined as the model predicted probability of an amino acid a position in the sequence, and the other is the semantic change of the mutation, defined as the $L_1$ distance between the embeddings of the mutated sequence and the wild-type sequence. In our experiment, we used the grammaticality of a mutant to correlate its fitness, as this was shown to outperform the prediction based on semantic changes in the CSCS study.

**Benchmarking experiments**. To assess ECNet's performance, we compared ECNet to other baselines using the original benchmark datasets that these methods were tested on in their publications. We ensured that the training and test sets are not overlapped in our experiments.

*Benchmarks on the Envision dataset*. We compared ECNet to the gradient boosting regression algorithm (denoted as 'Envision') proposed in the Envision dataset paper[35]. For each protein, we used 80% of the DMS data to train ECNet or other methods and the remaining 20% data to evaluate the model's performance. Spearman correlation was used as the evaluation metric. We used grid search to optimize the hyperparameters of ECNet. Note that Envision used 27 biological, structural, and physicochemical features to build the prediction model while our model only used the protein sequence to predict the functional measurements. To test the model's ability to identify variants that are more active than the wild-type sequence, we also converted the task into a classification problem, in which protein sequences with a function score greater than the wild-type sequence (with a function score 1) were labeled as positive samples, and the remaining sequences as negative samples. We used the AUROC score as the metric for this classification evaluation. We also compared ECNet to the Yang et al. (Doc2Vec) model on this dataset.

*Benchmarks on the DeepSequence dataset*. We compared ECNet to EVmutation, DeepSequence, and the Autoregressive model on the DeepSequence dataset that these methods have been tested on. The predictions made by these unsupervised approaches were collected from previous studies[15,46]. For ECNet, we performed five-fold cross-validation on this dataset and reported the average performance over all the five folds. Hyperparameters of ECNet were optimized using an inner-loop cross-validation. We used Spearman correlation as the evaluation metric. We also compared ECNet to supervised models that used global context representations (TAPE) or locally-fine-tuned global context representations (UniRep) on this dataset.

*Evaluation of variants prioritization*. We designed a simulation experiment of variants prioritization to assess how accurate and efficient ECNet is in retrieving high-performing variants[47]. We collected three large DMS datasets of proteins

avGFP[53], GB1[48], and Pab1[49], each with the fitness data of single and high-order mutants. We trained the model using 90% of the data (randomly sampled) and asked the model to predict the fitness for the remaining 10% data. For comparison, we ran One-hot encoder, Evmutation, and UniRep to predict the same test variants. We also included the ideal model, which used the ground-truth ranking of fitness values to rank variants, and the null model, which ranked variants with a random order, as references for the evaluation. For each protein, we repeated the experiments ten times. We calculated the recall as the evaluation metric, which is defined as the fraction of true top 100 variants the model recovered in its list of top $K$ predictions. This value of $K$ can be interpreted as the sequencing budget in actual experiments of directed evolution. We also used the maximum fitness (normalized by rank) observed in the top $K$ predictions as an additional metric to assess ECNet's ability in identifying the variants with the highest possible fitness. To demonstrate the efficiency of ECNet's prioritization, we computed its efficiency gain over random sampling as a function of budget $K$, which was quantified as the ratio between the recall of ECNet and the recall of the null model.

*Benchmarks of unsupervised ECNet*. We trained unsupervised ECNet models using the language model objective on the homologous sequences of three viral proteins, including influenza HA, HIV Env, and SARS-CoV-2 Spike, respectively. Hyperparameters of ECNet were optimized using an inner-loop cross-validation. The trained ECNet models were evaluated using fitness datasets for mutants of several proteins, including HA H1 WSN33, six HA H3 strains, BG505 and BF520 HIV Env, and SARS-CoV-2 Spike. The performance was evaluated using Spearman correlation between the output probability of a mutation given by unsupervised ECNet and the fitness score of that mutation. As our training objective followed that of CSCS, a protein language model developed to predict the escape of viral mutations, we compared our model to CSCS and validated that our model achieved a comparable performance as CSCS. For reference, we also trained a supervised ECNet on the viral proteins using five-fold cross-validation and found that the supervised model substantially improved the performance.

*Effects of training data size*. To investigate the effects of training data size, we randomly withheld 10% of data of each DMS study in the DeepSequence dataset as the test set. For the remaining 90% data, we trained two separate ECNet models by using all of them as training data (denoted as 100%) or randomly sampling 1/4 of them as training data (denoted as 25%). We also trained an unsupervised ECNet model without using any DMS data (denoted as 0%). The three models were all evaluated on the same test set.

*Comparison to randomly generated TEM-1 variants*. We trained an ECNet model and used it to predict the fitness of 146 TEM-1 inhibitor-resistant variants and a set of randomly generated variants. For every of the 146 TEM-1 inhibitor-resistant variants, we generate ten random variants that have the same mutated sites as the inhibitor-resistant variant but the alternative amino acid at each position is re-sampled uniformly from the 20-amino acid set. We ensured that the generated random variants do not overlap with any of the 146 variants.

**Prioritized high-order TEM-1 variants using ECNet**. We trained an ECNet model using DMS data of low-order TEM-1 variants and used the model to prioritize new high-order variants that were likely to have enhanced fitness. We sourced the training data from Firnberg et al.[55] and Gonzalez et al.[51], which measured fitness values of 98.2% (2536/2583) of all possible point mutants and 12.0% (12,374/102,855) of all possible consecutive double mutants, respectively. In both studies, the fitness of TEM-1 was defined as its resistance to ampicillin (Amp). We randomly sampled 95% of the combined datasets to form the training set and used the remains as the validation set. We used the default hyperparameters mentioned above for ECNet. The model was trained for 2000 epochs with early stopping if its performance on the validation set did not improve for 1000 epochs.

We used the trained model to predict the fitness of variants beyond single and consecutive double mutants and to identify new variants with improved fitness (as compared to the wild type). To reduce the exponential search space ($20^{286}$ sequences) of possible sequences, we focused on a restricted subspace where the mutations only occur on plausible function-related sites of TEM-1 documented in the literature. The detailed steps are as below. (1) We compiled a list of inhibitor-resistant TEM-1 variants supported by evidence in previous studies (see "Datasets"). The list contains 146 TEM-1 variants, each with 2–11 mutations with respect to TEM-1 wild-type sequence, covering 72 positions and 99 unique amino acid (AA) changes. (2) Based on each of the 146 variants, we generated new variants by considering all subset combinations of mutated positions of this variant and enumerating all AA changes that have appeared in these positions in the list. (3) From the newly generated sequences, we removed sequences that are identical to sequences in the 146-variant list, resulting in 18,937 sequences that form our restricted search space. We call these sequences "candidate variants". (4) We applied the trained ECNet model to predict the fitness for each of the candidate variants. Variants predicted to have lower fitness than the wild-type fitness were removed. (5) We used FoldX[38] to compute the change of structure stability for each variant. The PDB structure of wild-type TEM-1 was used as the template (PDB ID: 1XPB) and refined by the 'RepairPDB' function of FoldX. The 'BuildModel' procedure of FoldX was applied to mutate residues and compute the change of

stability. Variants with a large change of stability ($|\Delta\Delta G| > 3$ kcal/mol) were removed. (6) The remaining variants are sorted based on their predicted fitness, and we refer to those variants as prioritized variants.

To generate prioritized variants for experimental validations, we built two versions of ECNet models, a base version (ECNet-base) and an ensemble version (ECNet-ensemble). In ECNet-base, we trained three independent predictors using the default neural network architecture of ECNet, but each with a different loss objective, namely regression loss, classification loss, and regression loss with an auxiliary classification loss, respectively. The intersection of variants prioritized by the three predictors formed the final prioritized variants of ECNet-base. The use of three loss objectives here follows the intuition that the regression loss encourages the model to approximate the fitness of all variants and the classification loss focuses the model on identifying variants with fitness higher than the wild type. Combining the three predicted lists can prioritize more reliable predictions. We selected the top 28 variants in the intersected list for experimental validations. In ECNet-ensemble, we followed the same procedure as in ECNet-base but trained five replicates of predictive models for each loss objective. The predicted fitness of a variant was averaged over the five replicates. From the prioritized list by ECNet-ensemble, we selected 9 variants that were ranked at the top but different from what have been selected from the list predicted by ECNet-base.

In total, we used ECNet to identify 37 TEM-1 variants that were likely to demonstrate improved fitness as compared to the wild type. These variants contained two to six mutations (average 3.02 mutations per sequence), covering 22 positions in the TEM-1 sequence.

## Experimental validation of prioritized TEM-1 variants

*Materials and general methods.* Molecular biology reagents and chemicals were purchased from Fisher Scientific, Sigma-Aldrich, GOLDBIO, or New England Biolabs, Inc., unless specified otherwise. *Escherichia coli* DH5α (New England Biolabs, MA) was cultured in Luria–Bertani broth. DNA sequencing was performed at ACGT (Wheeling, IL). Primers were ordered from Integrated DNA Technologies (Coralville, IA) and listed in Supplementary Data 2. Plasmid pSkunk3-BLA was purchased from Addgene (plasmid 61531). PacBio Barcoded Universal Primers (Part Number: 101-629-100) was purchased from Pacific Biosciences (Menlo Park, CA).

*TEM-1 mutant creation.* TEM-1 mutants were constructed by overlapping PCR using primers (Supplementary Data 2) carrying targeted single or multiple mutation sites. Briefly, DNA fragments were PCR amplified by primers carrying targeted single or multiple mutation sites using pSkunk3-BLA plasmid as template and then gel purified. The purified DNA fragments were further fused by overlapping PCR to provide DNA fragments with complete TEM-1 gene fragments flanked by restriction enzyme (*Bam*HI and *Spe*I) digestion sites. After gel purification, the fused DNA fragments and pSkunk3-BLA plasmid were digested by *Bam*HI and *Spe*I. The digested TEM-1 gene with mutations and pSkunk3-BLA plasmid was gel purified and ligated by T4 DNA ligase and then transformed into *Escherichia coli* DH5α competent cells. The single colonies from the transformation plates were picked and cultured overnight at 37 °C. The mutation sites of mutants were confirmed by DNA sequencing using primers listed in Supplementary Data 2. In some cases, the constructed TEM-1 plasmids were used as PCR templates for creating other variants.

*Ampicillin resistance assay of TEM-1 mutants.* The plasmids harboring the genes encoding wild-type TEM-1, positive controls, and ECNet's predicted variants were mixed with equal concentrations and transformed into *Escherichia coli* DH5α competent cells (six replicates). After incubation at 37 °C overnight, the colonies from each of the transformation plates were immersed by ice-cold LB medium which were further scratched and pooled, yielding 10 ml cell suspension in LB medium. Inoculated 0.5 mL of the cell suspension into 50 mL LB medium supplemented with streptomycin (50 μg/ml) to allow the $OD_{600}$ of cultures to reach 0.5 with shaking at 37 °C. The cells were then washed twice by 1 volume of ice-cold 1X PBS and then resuspended in 1 volume of ice-cold 1X PBS. Finally, 100 μL cells were spread onto each of the freshly prepared LB agar plates with different concentrations of ampicillin (0, 300, 1500, and 3000 μg/ml), streptomycin (50 μg/ml), and IPTG (0.3 mM). The plates were incubated at 37 °C overnight. The colonies from each plate were then pooled and miniprepped to provide a plasmid mixture for each plate. The concentrations of the plasmids were determined by Qubit$^{TM}$ 4 Fluorometer (Invitrogen). PCR amplification of the targeted region with the same amount of total plasmids from each plate as templates by using target-specific primers tailed with a universal sequence (Supplementary Data 2) and Phusion Hot Start II High-Fidelity PCR Master Mix (Thermofisher, F-565S) was performed according to the manufacturer's instructions. The amplicons were then PCR barcoded by PacBio Barcoded Universal Primers using Phusion Hot Start II High-Fidelity PCR Master Mix. After gel purification, the barcoded amplicons were pooled with equal concentrations which were further used for SMRTbell library construction with the SMRTbell Express Template Prep Kit 2.0. PacBio sequencing with Sequel II System was then performed at the Roy J. Carver Biotechnology Center at University of Illinois at Urbana-Champaign. About two million reads with a mean read length of 949 bp were obtained from the sequencing. The reads were error corrected with circular consensus and demultiplexed. Further

bioinformatic analysis revealed read numbers of individual mutants from the corresponded plates with various concentrations of ampicillin. The fitness of each mutant at a certain ampicillin concentration was determined based on the ratio of the relative abundance of the mutant to wild-type TEM-1 in the plate with the related concentration of ampicillin and the relative abundance of the variant to wild-type TEM-1 in the plate without ampicillin.

*Fitness calculation.* The PacBio read data was processed using the TADA workflow (https://github.com/h3abionet/TADA). The fitness of a TEM-1 variant is determined by the ratio between the relative abundance of variants under the selection of a specific concentration and without selection. More precisely, for the fitness value $f_c(MT)$ of a variant MT at concentration $c$ ($c = 300$, 1500, or 3000 μg/mL) is calculated as

$$f_c(MT) = [N_c(MT)/N_c(WT)]/[N_0(MT)/N_0(WT)], \qquad (6)$$

where WT is the wild type and $N_c(\cdot)$ is the read count of a mutant under concentration $c$.

**Reporting summary.** Further information on research design is available in the Nature Research Reporting Summary linked to this article.

## Data availability
The following datasets generated or curated in previous publications were used: Envision dataset (https://doi.org/10.1016/j.cels.2017.11.003); DeepSequence dataset (https://doi.org/10.1038/s41592-018-0138-4); single and double mutants fitness (https://doi.org/10.1038/s41588-019-0432-9, https://doi.org/10.1016/j.jmb.2019.03.020, https://doi.org/10.1038/s41467-019-12101-z); TEM-1 single-mutation and double-mutation mutants fitness data (https://doi.org/10.1093/molbev/msu081, https://doi.org/10.1016/j.jmb.2019.03.020); high-order avGFP fitness (https://doi.org/10.1038/nature17995); inhibitor-resistant TEM-1 variants (downloaded from https://externalwebapps.lahey.org/studies/TEMTable.aspx and deposited at https://doi.org/10.6084/m9.figshare.16516608.v1); homologous sequences and fitness data of viral proteins (https://doi.org/10.1126/science.abd7331); PDB structure of TEM-1 (PDB ID: 1XPB). The fitness data of TEM-1 validation experiment is available in Supplementary Data 1.

## Code availability
The source code of ECNet is available at https://github.com/luoyunan/ECNet and on Zenodo[78] https://doi.org/10.5281/zenodo.5294461. ECNet was built on Python 3.7, PyTorch 1.4.0, Numpy 1.18.5, Scipy 1.4.1, Numba 0.45.1, Bio Python 1.78, SciKit-Learn 0.24.1, Pandas 1.2.3, msgpack-python 0.5.6, and TAPE 0.4.

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

## Acknowledgements

This work was supported by U.S. National Science Foundation under grant no. 2019897 (H.Z. and J.P.) and U.S. Department of Energy award DE-SC0018420 (H.Z. and J.P.). J.P. acknowledges the support from the Sloan Research Fellowship and the NSF CAREER Award. Y. Luo acknowledges the support from the CompGen Fellowship. BioRender.com was used to generate part of Fig. 5a.

## Author contributions

H.Z. and J.P. conceived and supervised the research project. Y.Luo. and J.P. developed the computational method. G.J. and H.Z. developed the validation pipeline. Y.Luo. implemented the software and performed the computational experiments. Y.Luo analyzed the benchmarking results of the computational model with support from T.Y., Y.Liu. and L.V. G.J. performed the experimental validation task with support from T.Y. G.J. and Y.Luo. analyzed the validation results. H.D., Y.S. and W.W.Q. curated the data used to develop the computational method. Y.Luo., G.J., H.Z. and J.P. wrote the manuscript with support from all authors.

## Competing interests

The authors declare no competing interests.
