## [Peer Review File · Nature Communications]

ECNet: an evolutionary context-integrated deep learning framework for protein engineeringReviewers' Comments:

Reviewer #1:

Remarks to the Author:

This study presents ECNet (evolutionary context-integrated neural network), a deep learning model that predicts protein fitness from sequence in order to guide protein engineering.

The tool innovation was built on a sound rationale. Compared to earlier efforts, ECNet combines fitness signals arising from both global contexts (extracted from protein language models for the protein universe) and local contexts (extracted from co-evolution analysis for homologous proteins) so that better specificity to the proteins to engineer can be achieved. The two types of protein contexts or representations were concatenated and fed to LSTM, a recurrent neural network, and fully connected layers for supervised sequence-to-function prediction or unsupervised sequence models.

The numerical experiments were very comprehensive and insightful. After demonstrating the relevance of co-evolutional signals to double mutation effects, the authors compared ECNet with Doc2Vec (global contexts only from language models) and Envision (with man-made features) on a few datasets and demonstrated the potential benefit of using local co-evolution contexts. The authors then compared ECNet favorably to three unsupervised and two supervised methods on a larger dataset of deep mutation scanning experiments. They went on with experiments to address other practical concerns, such as predicting epistasis, predicting higher-order mutational effects based on lower-order measurements, and unsupervised ECNet when no measurements are available.

ECNet was used to engineer 37 predicted ampicillin-resistant variants of TEM-1 beta-lactamase that were experimentally validated to contain 50%-90% true positives at various ampicillin concentrations. Some of the variants were found to be even more resistant than the best training example (positive control). Importantly, these variants cover high-order variants combining non-consecutive pre-selected sites.

Please find below some comments which might help a more clear and more convincing presentation.

Major:

1. One major claim of ECNet is that the use of local evolutionary contexts in homologous proteins helps the specificity of protein engineering. This was indeed demonstrated when comparing ECNet to unsupervised EVMutation, DeepSequence, and Autoregressive, or supervised TAPE and UniRep, all of which are using global contexts from the protein/domain universe.

What remains to be shown is whether the co-evolutionary signals from direct coupling analysis is convincingly better than other baselines for local contexts. For example, all aforementioned 5 methods can be easily fine-tuned in unsupervised learning for the target protein family so that local evolutionary contexts can be introduced. Could these baselines for specificity in protein engineering be compared to ECNet? I suspect that, compared to those methods using target-specific albeit alignment-free sequences, ECNet using MSA-based co-evolution signals should still show improvements, which would make the study even more convincing.

2. Figure 3 is very clear and convincing, which begs for more insights. What kinds of datasets / protein families do ECNet perform better or worse in absolute performance (ρ)? What kinds do ECNet gain more improvement relative to the other methods? Apparently the accuracy of local co-evolutional signals from direct coupling analysis is impacted by the number of sequences or effective sequences in

the MSA for the given query protein. But small size of MSA (1K) did not lead to particularly worse relative performance, as seen in Figure 3c.

3. The training/test variant split was often in a random way or controlled based on the order (lower-to-higher order prediction). Could any experiments be done to examine the generalizability to new sites? In other words, were there any simulation experiments evaluated based on test variation sites excluded in the training variation sites? In the beta-lactamase experiment, was there any of the 37 variants involving mutation sites outside the training variants' mutation sites?

4. Figure 4C: how was random variants generated? By "random", did the authors mean uniform distribution of 20 amino acids at each position or the learned family-specific distribution? Figure 4D: why not compare to the better performing supervised TAPE?

5. What exactly was the classification loss? The sum of cross entropy for 10 independent binary classifications, a loss for ordered or non-ordered multi-class classification, or something else?

6. Was the unsupervised ECNet trained with a cold start or a warm start (e.g. using pretrained TAPE)? Was it trained on domain sequences or protein sequences of the target family? (for instance, for spike proteins was it trained on RBD alone or the whole sequence?) When demonstrating unsupervised ECNet on 3 viral proteins, how many sequences were used for each protein and were there enough sequences for unsupervised training with a cold start?

7. When prioritizing high-order TEM-1 variants for ampicillin resistance, how important was it to restrict the variation space to just combinations of training variation sites? How important was it to filter variants based on FoldX-predicted stability? Any computational or/and experimental way to quantify such ablation effects?

8. Any justification of the 3 ampicillin concentration levels? How to interpret the various hit rates at various concentration levels? If the hit rates increase with concentration levels, is there any "control" hit rate to be compared to in Figure 5C, so that the reported hit rates of ECNet at various concentration levels can be better interpreted?

Minor:

"BARD1 biding" (binding?) appeared twice in Figure 2 and once in Figure S1. "Variants construction missing a "t" in Figure 5a.

Reviewer #2:

Remarks to the Author:

The authors present a learned model that combines global embeddings of protein sequences with an embedding of the Multiple Sequence Alignment (MSA) of a protein. They apply their model to predict the fitness of variants in a large number of published Deep Mutational Scanning (DMS) datasets, as well as some novel b-lactamase variants, and show that their model achieves marginally better performance than existing methods.

Some similar papers include:

- Unified rational protein engineering with sequence-based deep representation learning, 2019, Alley. Presents the idea that embedded protein representations could be valuable for predicting experimental datasets, and benchmarks them against several DMS datasets.
- Evaluating Protein Transfer Learning with TAPE, 2019, Rao. Applies language models to a variety of prediction tasks, including DMS dataset prediction.
- Low-N protein engineering with data-efficient deep learning, 2021, Biswas. This paper expands upon this idea, and applies these techniques to engineer an evolved GFP and also a TEM-1 β -lactamase

The main contribution of this paper is a novel encoding that combines a global language model representation with an encoding of the protein's MSA. Adding information about homologous sequences of a protein improves the predictive capabilities of the model.

Overall, I doubt that this paper will have a very large impact on the field. The improvement in performance is marginal, and comes at the cost of restricting the model to work well only for proteins for which excellent MSAs can be obtained. The main result - that adding in MSA information improves predictive capabilities - is not a surprise: the highly related field of machine learning-based protein structure prediction has observed this numerous times. The conceptual contribution of proposing that protein embeddings can be used to predict experimental datasets has been made already in numerous papers, and there have already been many iterations on exactly how the embedding should be constructed (several of which they cite, such as TAPE). This paper even uses the same protein evolution test case as a previous paper: β -lactamase evolution. Overall, it is a nicely put together manuscript, but does not offer very strong conceptual or methodological advances.

Some things that would strengthen the paper:

- It would be helpful to do analysis on your own model to confirm that the improved performance is because of the MSA. There's a section on ablation analysis in figure S2, could you expand upon this somewhat? From this graph, it seems that the local evolution representation alone performs on-par with ECNet. It is unclear to me whether this is the analysis I'm looking for.
- How does performance depend on the number of sequences in the MSA and the quality of the MSA? You have some data about this in figure 1b, but it would be nice to see more nuanced analysis.
- How do you determine the number of parameters in the top model? Do you adjust the number of parameters in response to the size of the experimental dataset?
- How does this model respond to noisy experimental data? If you add x% noise to the training data on one of the datasets, how does it impact the spearman correlation? We might expect that some models are more brittle/robust to experimental data noise - this would be a valuable thing to measure and I'd be interested to see if your model out-performs others.
- In Figure S3, it would be great to add the performance of a simple one-hot encoded model.

Reviewer #3:

Remarks to the Author:

In this work, the authors developed one machine learning-based evolutionary context-integrated neural network (ECNet) technique to exploit evolutionary contexts to predict functional fitness in protein engineering. The algorithm integrates the evolutionary information from the homologous sequences and the evolutionary context from the enormous protein sequence universe to map the sequence to the function of the protein mutants. The authors claim the superior performance of ECNet compared to existing machine learning algorithms by using ~ 50 deep mutagenesis scanning and random mutagenesis datasets. Also, the authors validated their method experimentally on identifying the variants of TEM-1 β -lactamase through the engineering process.

While the manuscript is well presented but few portions need more clarification.

1. The strength of pairwise residue dependencies does not correlate well with the fitness of double mutants (Figure 1A). Still, authors rely on this correlation to build their method. A justification for using this low correlation is required.
2. Despite the dataset section, it is not clear what is the training data, what is the testing data, what fraction of the data is a positive sample, what fraction is a negative sample. Also, it is not clear whether the dataset is balanced or not. Since any ML-based or DL-based technique heavily depends on the dataset, so more details are required for the dataset.
3. The authors mentioned utilizing different datasets from different published works. However, it is not clear in total how much data is utilizing for ECNet. There is a possibility that the dataset may have some similarities. Did the author check for data duplicates or data redundancy? An explanation is required in this context.
4. The authors have used a language model for protein sequence. Despite the results, an analysis (or explanation) is required regarding the validation of such use since it is the core part of their ML-based method. The authors may include this in the introduction part.
5. Authors mentioned that "Using a large corpus of protein sequences such as UniProt and Pfam, a language model learns to predict the next amino acid given all its preceding amino acids." Again a justification is required since one or a few critical mutations may destroy the protein structure.
6. The authors evaluated the performance of their model using 5-fold cross-validation (Figure 2). How many random experiments were performed? It would be good to iterate it for 100 runs and present the mean and standard deviation of the runs. Apart from relative improvement (Figure 2b), please provide the absolute values of AUROC.
7. "As protein engineering focuses on identifying variants with improved properties than the wild type, we further evaluated the model performance using a classification metric (AUROC score), in which variants with higher function measurements than the wild-type sequence are defined as positive samples and the remaining variants as negative samples." How relevant is it in differentiating positive and negative samples? What is the threshold value used?
8. While mapping the sequence to function, whether authors consider the stability of the protein? How do they are certain about stability? Specifically, when authors comment that "to assist protein engineering, the machine learning model needs to prioritize variants that are not only structurally stable and non-deleterious but also with enhanced properties."
9. In the dataset section, authors are requested to differentiate training set from the testing set and positive data from negative data - specifically for viral proteins.
10. On page 21, what is the difference between N and L? Does $L + (\#gaps) = N$? Please add few lines for describing L more clearly.
11. How do you achieve $O(NL^2)$? Few lines on the derivation will be good.
12. In the dimensionality reduction step, how do you decide on d? What is the value for d? Is it protein sequence invariant?
13. Please draw the architecture diagram for Sequence-to-function neural network model. Figure 1C is not enough.
14. Does ECNet capable of predicting multiple functions? Is there any possibility?
15. Do the authors analyze what fraction of engineered cases retained the same function, and what fraction changes the function?
16. Please provide more details on hyperparameter optimization of ECNet.
17. On page 23, "The batch size was set to 128 and the maximum number of training epochs was set to 1,000 with an early stop if the performance has not been improved for 500 epochs." And in page 28, "The model was trained for 2,000 epochs with early stopping if its performance on the validation set did not improve for 1,000 epochs." Are they two different models? How many different models are there? It will be good if authors provide detailed data (maybe tabular or so) describing how many models are there and which model uses what sort of features, data and comparing with which existing methods.

Some minor suggestions to improve the readability of the manuscript-

18. Protein engineering involves mutation and InDel operations on protein. Although, the authors focused on mutation only and mostly focused on sequence-to-function mapping but "Protein

engineering" as the title of the manuscript is a bit misleading. Authors may consider making it more specific.

19. It is not clear from the main text regarding the measure of "16%-60%" improvement.

20. Page 27, first line – is there a typo? Please check.

21. It will be good if few lines can be added to emphasize more on the protein language model.

22. Please use the equation number wherever required. It will be convenient for the reader.

We thank our reviewers for their careful reading and critical feedback. We appreciate their insightful comments and constructive suggestions, which are helpful for improving the quality of our manuscript. In the revised manuscript, we have implemented our reviewers' comments. In particular, we have made the following major changes:

Relationship between performance and MSA. To provide more insights on our model's performance improvement, we have performed several computational experiments and thoroughly analyzed the relationship between our model's performance and input data characteristics. We also expanded the analysis to the relationship between the model's performance and DMS data completeness. Our results showed that ECNet's performance was correlated with the number of sequences in MSA and the completeness of the training DMS data (**Figures S8-S9**, our responses to **Reviewer 1: Q3** and **Reviewer 2 Q2**).

Model generalizability. We designed new experiments to directly assess the model's generalizability, including the generalization to new protein sites and new amino acids (**Figures S10-S11**, our response to **Reviewer 1: Q3**). We found that ECNet had better generalizability than the DeepSequence model.

Model robustness. We added a new simulation to assess ECNet's robustness against noise in training data (**Figure S4**, our response to **Reviewer 2: Q4**). We observed ECNet was very robust against several noise levels.

Ablation studies. We performed ablation experiments to study the contributions of individual features (local and global evolutionary features) to the prediction performance using both single-mutation and double-mutation fitness data (**Figure S3**, our response to **Reviewer 2: Q1**). Our results showed that both the global and local evolutionary features provided additional values and improved the prediction performance in most cases.

Methodology details. We substantially revised our manuscript and provided the details about hyperparameter choices and train/test data split (see our responses to **Reviewer 2: Q3** and **Reviewer 3: Q2, Q9, Q12, Q16**). We also added more explanations and justifications in our manuscript to improve the clarity and readability of our manuscript (see our responses to **Reviewer 1: Q6, Q7, Q8** and **Reviewer 3: Q1, Q4, Q5, Q7**).

Our point-by-point response to our reviewers' comments can be found below. The reviewers' comments are reproduced in **blue text** and our responses are in **black text**. The texts copied and pasted from our revised manuscript are shown in **red text**.

Reviewer #1

This study presents ECNet (evolutionary context-integrated neural network), a deep learning model that predicts protein fitness from sequence in order to guide protein engineering.

The tool innovation was built on a sound rationale. Compared to earlier efforts, ECNet combines fitness signals arising from both global contexts (extracted from protein language models for the protein universe) and local contexts (extracted from co-evolution analysis for homologous proteins) so that better specificity to the proteins to engineer can be achieved. The two types of protein contexts or representations were concatenated and fed to LSTM, a recurrent neural network, and fully connected layers for supervised sequence-to-function prediction or unsupervised sequence models.

The numerical experiments were very comprehensive and insightful. After demonstrating the relevance of co-evolutional signals to double mutation effects, the authors compared ECNet with Doc2Vec (global contexts only from language models) and Envision (with man-made features) on a few datasets and demonstrated the potential benefit of using local co-evolution contexts. The authors then compared ECNet favorably to three unsupervised and two supervised methods on a larger dataset of deep mutation scanning experiments. They went on with experiments to address other practical concerns, such as predicting epistasis, predicting higher-order mutational effects based on lower-order measurements, and unsupervised ECNet when no measurements are available.

ECNet was used to engineer 37 predicted ampicillin-resistant variants of TEM-1 beta-lactamase that were experimentally validated to contain 50%-90% true positives at various ampicillin concentrations. Some of the variants were found to be even more resistant than the best training example (positive control). Importantly, these variants cover high-order variants combining non-consecutive pre-selected sites.

Please find below some comments which might help a more clear and more convincing presentation.

Response: We thank the reviewer for a nice summary of our work.

Major:

1. One major claim of ECNet is that the use of local evolutionary contexts in homologous proteins helps the specificity of protein engineering. This was indeed demonstrated when comparing ECNet to unsupervised EVMutation, DeepSequence, and Autoregressive, or supervised TAPE and UniRep, all of which are using global contexts from the protein/domain universe.

What remains to be shown is whether the co-evolutionary signals from direct coupling analysis is convincingly better than other baselines for local contexts. For example, all aforementioned 5 methods can be easily fine-tuned in unsupervised learning for the target protein family so that local evolutionary contexts can be introduced. Could these baselines for specificity in protein engineering be compared to ECNet? I suspect that, compared to those methods using target-specific albeit alignment-free sequences,

ECNet using MSA-based co-evolution signals should still show improvements, which would make the study even more convincing.

Response: We sincerely thank the reviewer's comments and suggestions. We would like to clarify that the three unsupervised methods, EVmutation, DeepSequence, and Autoregressive, have already been trained on local contexts by default (instead of global protein universe) according to the method descriptions in those papers. For example, both EVmutation and DeepSequence were trained on the multiple sequence alignment (MSA) of homologous sequences of the target protein, while Autoregressive was also trained on alignment-free homologous sequences. Therefore, those three methods have already captured the local evolutionary contexts.

The other two supervised methods TAPE and UniRep, as the reviewer noted, are trained on global contexts from general protein sequences (e.g., Pfam or UniProt) and can be fine-tuned on the homologous sequences of the target protein. In fact, in the UniRep paper, the authors have introduced a trick called 'evotuning', in which the UniRep model was first trained on 24 million UniRef50 sequences and then fine-tuned on tens of thousands of homologous sequences of the target protein. Similarly, TAPE can also be fine-tuned in this way. In our benchmark experiments, the UniRep model we compared to has already been evotuned (see "Baseline methods" in the Methods section). Therefore, the UniRep results in **Figure 3b** and **Figure S4** of our manuscript are the comparisons that the reviewer was looking for. We observed that ECNet still outperformed the UniRep model that has been fine-tuned on local contexts.

2. Figure 3 is very clear and convincing, which begs for more insights. What kinds of datasets / protein families do ECNet perform better or worse in absolute performance (ρ)? What kinds do ECNet gain more improvement relative to the other methods? Apparently the accuracy of local co-evolutional signals from direct coupling analysis is impacted by the number of sequences or effective sequences in the MSA for the given query protein. But small size of MSA (1K) did not lead to particularly worse relative performance, as seen in Figure 3c.

Response: We thank the reviewer's suggestion. In our revision, we have performed several analyses to investigate the relationship between ECNet's prediction performance and the properties of MSA data and DMS (deep mutational scanning) data. Overall, (i) at the protein level, we found that ECNet's prediction performance is better for proteins with more homologous sequences; (ii) at the residue level within a protein, we found that ECNet predicts fitness better for sites that are more conserved; and (iii) by comparing ECNet with DeepSequence, we additionally found that ECNet gained more improvements if the training DMS dataset is more complete. We provide our detailed results below.

Prediction performance correlates with the number of MSA sequences. First, we expanded the results in **Figure 1b**, where we used the residue co-variation derived from MSA to predict the fitness of single-mutation variants of proteins in the Envision dataset. Recall that in this experiment, a direct coupling analysis model was used to fit the MSA data and the change of the co-variation terms upon mutations was used to predict the fitness data (i.e., in an unsupervised manner). Here, we replotted the

results in a scatter plot in **Figure R1**. It is clear that the number of MSA sequences correlated very well with the prediction performance (Spearman's $\rho=0.82$) because more sequences in the MSA gave more accurate and robust estimations about the amino acid preference at each site in the sequence.

Next, we extended the analysis to the benchmark results on the DeepSequence dataset, which consists of DMS data of 38 proteins. We found that ECNet's prediction performance still positively correlated with the number of sequences in the MSA (**Figure R2a**; Spearman's $\rho=0.27$). The correlation coefficient was lower than our analysis above for the Envision dataset in **Figure R1**. One major reason is that, unlike the unsupervised approach in **Figure R1** that entirely relied on MSA to predict fitness, here our ECNet algorithm is a supervised model and utilized both MSA data and training fitness data to predict fitness. Therefore, for some proteins, ECNet is still able to predict fitness rather accurately even though that protein only has very few homologous sequences in MSA. For example, ECNet achieved a correlation >0.8 for several proteins with less than 10k MSA sequences (**Figure R2a**).

Less-conserved sites are easier to predict. Motivated by the reviewer's comments, we further investigated what other factors might impact ECNet's performance. Proteins do not mutate in a uniform way in the evolution, i.e., some sites in the sequence are more conserved towards specific amino acids (AAs) while others can tolerate different AAs. Therefore, we hypothesized that the mutation effects at

different sites are not equally ‘hard’ to predict. We thus plotted the relationship between the entropy at a site in the sequence and ECNet’s prediction accuracy for that site (**Figure R2b**). The entropy value is calculated using the AA frequency at each site (column) in the MSA and quantifies the conserveness of each site -- a lower entropy means this site has a dominant preference towards specific AAs while a higher entropy indicates that many AAs are likely to appear at this position (in terms of occurrence frequency). We observed that conserved sites (lower entropies) are easier to predict at higher correlations (**Figure R2b**; $\rho=-0.34$). This is because, for conserved sites, the mutation (no matter what AA it mutates to) is likely to be damaging, which is more straightforward for the model to predict. In contrast, for tolerable sites, the mutation consequence can be damaging, neutral, or even beneficial, raising more ambiguity for the predictive model.

completeness is defined as the percentage of screened variants among all possible single-mutation variants for a protein, i.e., $M/(19*L)$, where M is the number of screened variants in the DMS data and L is the length of the protein sequence. Each point represents one protein in the DeepSequence dataset. ECNet's prediction performance is summarized by the average of five-fold cross-validation. **(d)** Same as **(c)** except that the DMS completeness is defined for each site. The per-site completeness is defined as the percentage of screened mutations among all possible single-mutation mutations for a position, i.e., $m/19$, where m is the number of screened mutations for this position in the DMS data. Each point represents one protein in the DeepSequence dataset. The per-site completeness is first averaged over all test sites in a fold and then averaged over all five folds. (MSA: multiple sequence alignment; DMS: deep mutational scanning.) This figure is labeled as **Figure S9** in the revised manuscript.

DMS data completeness influences model accuracy. Finally, we analyzed for which proteins has ECNet gained more improvements compared to DeepSequence, an existing unsupervised method. As ECNet is a supervised algorithm, we hypothesized that its prediction accuracy is impacted by the quality of training data. Here, we analyzed the relationship between the model's prediction performance and the completeness of a DMS dataset. A protein with sequence length L has $19*L$ possible single-mutation variants. However, in practice, a DMS study may not always be able to sample all the possible single-mutation variants due to technical reasons. Using the single-mutation DMS data of 37 proteins from the DeepSequence dataset, we plotted the completeness of the DMS data and ECNet's prediction performance in a scatter plot (**Figure R2c-d**). For both sequence-level and per-site completeness (see **Figure R2** legend for definitions), we observed that ECNet trained on more complete DMS data has achieved better results. As a more complete DMS dataset samples more variants, either covering more sites or screening more mutations on the same site, the ECNet models trained on this dataset are thus more accurate for a wide range of sites and mutations. A recent concurrent study also obtained a similar observation (Wittmann et al, *bioRxiv*, 2020; DOI:10.1101/2020.12.04.408955).

To conclude, our new analyses here dissected the factors that impact ECNet's prediction performance. We found that both MSA characteristics and DMS data properties can impact the prediction accuracy. We added the result figures to the Supplementary Information and the corresponding text to the Discussion section in our manuscript, which is also reproduced in the box below.

Revised text (Section "Discussion"):

ECNet's prediction performance is impacted by the MSA characteristics and DMS data properties. For example, ECNet predicts better for proteins with more homologous sequences (**Figure S8 and Figure S9a**), for sites that are more conserved within a protein (**Figure S9b**), and for proteins that have a more complete DMS dataset (**Figure S9c-d**).

References:

Wittmann, Bruce J., Yisong Yue, and Frances H. Arnold. "Machine learning-assisted directed evolution navigates a combinatorial epistatic fitness landscape with minimal screening burden." *bioRxiv*, (2020).

3. The training/test variant split was often in a random way or controlled based on the order (lower-to-higher order prediction). Could any experiments be done to examine the generalizability to new sites? In other words, were there any simulation experiments evaluated based on test variation sites excluded in the training variation sites? In the beta-lactamase experiment, was there any of the 37 variants involving mutation sites outside the training variants' mutation sites?

Response: We first answer the last question. For the beta-lactamase experiment, as our machine learning was trained on a DMS dataset that measures the fitness for almost all single-mutation TEM-1 variants (2,536/2,583=98.2%), the mutation sites of our 37 engineered variants are all covered by training variants' mutation sites. But note that our 37 engineered variants are higher-order combinatorial mutants and they are not identical to any of the training variants.

We thank the reviewer for the suggestion on examining the generalizability of ECNet to new sites. It is important to evaluate this kind of prediction performance because it not only assesses the model's generalizability but also would reveal insights on which mutations should be experimentally tested to empower the machine learning model. We thus added several new experiments and analyses in our revised manuscript. In short, we first performed a site-wise train/test split (as suggested by the reviewer) to assess the generalizability of ECNet. We found that ECNet still outperforms a baseline model (DeepSequence) under this setting, although its performance has decreased compared to that under a random split setting or an AA-wise split setting. Inspired by the reviewer's question here, we also conducted additional experiments to study a natural follow-up question, i.e., which mutations should be tested in the DMS experiment, especially when the experimental testing budget is limited, to build a better machine learning model? In this regard, we examined the exploration-exploitation tradeoff in selecting mutants to test. Our results are detailed below.

Site-wise split is challenging for machine learning models. First, as suggested by the reviewer, we performed a new cross-validation experiment where we split the training and test sets based on the position (site) of the protein sequence (called "site-wise split", see **Figure R3a** for a schematic visualization). More specifically, for a DMS dataset, we randomly sampled 80% of sites of the protein sequences and used the mutation data on those sites as training data. The model was then tested on the mutation data of the remaining 20% of sites. We tested this split strategy on the 37 single-mutation DMS datasets curated in the DeepSequence dataset (**Figure R3c**). For reference, we also showed the performance of DeepSequence. We observed that under this site-controlled split setting, although still outperforming DeepSequence on most of the proteins, ECNet's prediction performance has decreased compared to that under a random split setting (see **Figure 3a** in the manuscript for the random-split performance). In particular, ECNet has had a better Spearman's correlation on 34/37 proteins in the random split experiment, while only improved for 20/37 proteins in the site-wise split experiment here. The performance losses were not unexpected and have also been observed in a recent study (Shamsi et al., *The Journal of Physical Chemistry*, 2020; DOI: 10.1021/acs.jpcc.0c00197). This result suggests that the mutation effects of sites far away from each other may not correlate with mutations in other sites, partially due to the different evolutionary conserveness or structural importance at each site. The

implication of this result is that, ideally, the machine learning model should be trained on DMS data for a large range of sites so that it would be able to predict more accurately for more sites in the sequence.

Figure R3. Prediction performance on sequence site-wise and amino acid-wise (AA-wise) train/test data split. (a-b) Schematic visualizations of train/test split. A DMS dataset is visualized as a matrix where the x-axis represents the index of position in the sequence and the y-axis represents the AA types the site mutates to. To split strategies are considered: (a) Site-wise split: the deep mutational scanning (DMS) dataset is split based on the sequence position (site) in the sequence. Mutants in 80% of the sites are randomly sampled as training data and mutants in the remaining sites are used as test data. The partition of train/test sites is only for the schematic visualization purpose. The actual training sites are randomly sampled and not necessarily the 80% leftmost sites; (b) AA-wise split: the DMS dataset is split based on AA types of mutations. For each site, 80% of the mutations are randomly

sampled and added to the training set and the remaining 20% mutations are added to the test set. The partition of train/test AA types is only for the schematic visualization purpose. The actual training mutations are randomly sampled and not necessarily the first 80% alphabetically-ordered AA types. **(c-d)** Comparison of ECNet to DeepSequence on 37 single-mutation DMS datasets for both **(c)** site-wise split and **(d)** AA-wise split settings. The DeepSequence's performances are the same in **(c)** and **(d)** as it is an unsupervised model. (AA: amino acid; DMS: deep mutational scanning.) This figure is labeled as **Figure S10** in the revised manuscript.

Increased site coverage of training data improves model performance. The above result led us to a second train/test split strategy that we called "AA-wise split", in which we split the data along the amino acid dimension (**Figure R3b**). More specifically, for each site of the sequence, we randomly sampled 80% of mutations measured on this site and added them to the training set, and the remaining 20% added to the test set. Under this setting, we ensured that the training set covered some DMS data for every site in the sequence. We found the ECNet model trained on this data achieved better performance compared to that of the above site-wise split experiment (**Figure R3d**). It has outperformed DeepSequence's Spearman correlation on 35/37 proteins. (Note that DeepSequence is an unsupervised model so its performance remained the same in **Figures R3c-d** regardless of the split strategy.) ECNet's improvement over DeepSequence (median $\Delta\rho=0.225$) was even slightly higher than the improvement achieved under the random split setting (median $\Delta\rho=0.196$). The result here suggested that in most cases ensuring a larger site coverage in the training data can improve the model prediction performance.

Exploration-exploitation tradeoff under limited experimental budget. Inspired by the reviewer's question and the above results, we then conducted experiments to study a natural follow-up question that has important implications for DMS experimental design. That is, given a limited experimental budget, should we test a few sites but screen more mutations for each site thoroughly (i.e., exploit more), or test just a few mutations for each site but cover more sites (i.e., explore more)? Which one would lead to a better ML model? We thus controlled the total number of training variants (fixed test budget) but sampled them from 50%, 75%, 100% sites of the sequence, respectively, and sample test variants uniformly from along the sequence site dimension (**Figure R4a**; see the legend for the details). These three datasets represented different levels of the exploration-exploitation tradeoff. We found that the test correlation increased as the exploration ratio increased and that the model with the most exploration achieved the best correlation on the test set (**Figures R4b-c**). The result suggested that if we do not have a prior that functional mutations are only enriched for a few sites, screening more sites (even only a few mutants at each site) would help the ML model better capture the mutation effects and increase the likelihood of discovering better variants.

test budget (i.e., the area of the purple region remains the same). **(b)** ECNet's performance is assessed using 37 single-mutation DMS datasets curated in the DeepSequence dataset. The box plot shows ECNet's test performance for exploration ratios 0.5, 0.75, and 1.0. The performance of unsupervised ECNet is also shown for reference. Each point represents the correlation on a single-mutation DMS dataset. The Spearman's correlation increases as the exploration ratio r increases. **(c)** Pairwise performance comparison between ECNet at different exploration ratios and the unsupervised ECNet. The underlying correlation data is the same as in **(b)**. Each point represents a single-mutation DMS dataset from the DeepSequence dataset. The improvements (mean $\Delta\rho$) achieved by supervised ECNet over the unsupervised ECNet are increasing as the exploration ratio r increases. This figure is labeled as **Figure S11** in the revised manuscript.

In conclusion, our new experiments revealed that ECNet still outperforms the baseline model (DeepSequence) even for new sites it has not been trained on. However, this was a more challenging scenario for ECNet and its performance has dropped compared to the version where it was trained on randomly sampled data. This result also suggested the importance of training data design when developing ML models for protein engineering and the tradeoff between exploration and exploitation is an important factor to consider if the budget is limited. We added the resulting figures to the Supplementary Information and the corresponding text to the Discussion section of our manuscript, which is also reproduced in the box below.

Revised text (Section "Discussion"):

Our additional tests used both a sequence site-wise strategy and a per-site AA-wise train/test split strategy to assess ECNet's generalizability. Despite the challenging setting, ECNet still outperformed DeepSequence when predicting for new mutation sites (**Figure S10**). Further tests investigation found that the exploration-exploitation tradeoff of training data also influenced the model performance (**Figure S11**). This implies that the design of more effective training data should be taken into account when developing ML algorithms to assist protein engineering, especially when the experimental test budget is limited.

Reference:

Shamsi, Zahra, Matthew Chan, and Diwakar Shukla. "TLmutation: predicting the effects of mutations using transfer learning." *The Journal of Physical Chemistry B* 124.19 (2020): 3845-3854.

4. Figure 4C: how was random variants generated? By "random", did the authors mean uniform distribution of 20 amino acids at each position or the learned family-specific distribution? Figure 4D: why not compare to the better performing supervised TAPE?

Response: In **Figure 4c** of our manuscript, we wanted to compare the predicted fitness of the 146 TEM-1 inhibitor-resistant variants and a set of randomly generated variants. The random variants were generated as follows: For every of the 146 TEM-1 inhibitor-resistant variants, we generated ten random

variants that have the same mutated sites as the inhibitor-resistant variant but the alternative amino acid at each position is re-sampled uniformly from the 20-amino acid set. We have added these details in the “Methods - Benchmarking experiments” section.

In **Figure 4d** of our revised manuscript, we have compared it to TAPE. We observed that TAPE more accurately predicted the epistasis than EVmutation on the six proteins. Our ECNet still outperformed or achieved comparable performance as TAPE. The figure is reproduced below in **Figure R5**.

Figure R5. Spearman correlation of experimentally measured epistasis and predicted epistasis. The Spearman correlations achieved by ECNet were comparable or significantly higher than that of EVmutation and TAPE for all six proteins (one-sided rank-sum test $P < 10^{-4}$). This figure is labeled as **Figure 4d** in the revised manuscript.

5. What exactly was the classification loss? The sum of cross entropy for 10 independent binary classifications, a loss for ordered or non-ordered multi-class classification, or something else?

Response: The classification loss is essentially the cross-entropy loss of a (non-ordered) ten-class classification. We have clarified this in our revised manuscript. We also thank the reviewer for mentioning the ordered loss. This is a direction that one can explore more to improve the robustness of the model, for example, implementing the objective function as an ordinal regression or ordinal classification.

6. Was the unsupervised ECNet trained with a cold start or a warm start (e.g. using pretrained TAPE)? Was it trained on domain sequences or protein sequences of the target family? (for instance, for spike proteins was it trained on RBD alone or the whole sequence?) When demonstrating unsupervised ECNet on 3 viral proteins, how many sequences were used for each protein and were there enough sequences for unsupervised training with a cold start?

Response: The unsupervised ECNet was trained with a cold start in our work as we found that the effective size of training data of those viral proteins is sufficient to train a model (as detailed below). However, when the target family data is limited, the model could be beneficial to train it with a warm start, e.g., by initializing the parameters of ECNet using the embedding obtained from TAPE.

The unsupervised ECNet is trained on the whole protein sequences of the target family, rather than on the domain sequence only. This choice followed the same setting in previous work (Hie, et al. *Science*, 2021; DOI: 10.1126/science.abd7331).

For the training on the 3 viral proteins, we used 44,851 unique influenza A hemagglutinin (HA) protein sequences observed in animal hosts, 57,730 unique HIV-1 Env protein sequences, and 4,172 unique Spike and homologous protein sequences. Note that in the unsupervised training, one training data point corresponds to one masked amino acid in a sequence. Therefore, the effective training size is (the number of sequences * the number of AAs in the sequence), which is much larger than the number of sequences. For example, the average length of HA sequence is 856 amino acids, so the effective size of training data points is $856 * 44,851 \sim 38$ million. We found that this magnitude of training data size is enough for the model to achieve a good accuracy when trained with a cold start. When data is very limited, we can train the model with a warm start, as the reviewer noted.

References:

Hie, Brian, Ellen D. Zhong, Bonnie Berger, and Bryan Bryson. "Learning the language of viral evolution and escape." *Science* 371, no. 6526 (2021): 284-288.

7. When prioritizing high-order TEM-1 variants for ampicillin resistance, how important was it to restrict the variation space to just combinations of training variation sites? How important was it to filter variants based on FoldX-predicted stability? Any computational or/and experimental way to quantify such ablation effects?

Response: We want to first clarify that the variation space in the TEM-1 prioritization experiment is not the combination of variation sites of our training data (single-mutation and consecutive-double-mutation variants). Rather, they are the combinations of mutated sites of 147 inhibitor-resistant TEM-1 variants curated in the literature ("known good variants"). We did not use those 147 variants in our training data. Regarding the reviewer's question, we think the restriction of variation space and the FoldX-based filtering would not have a strong effect on how good are the variants that ECNet can discover, yet implementing those two steps does have advantages in terms of efficiency, as discussed below.

We think restricting the variation space to the combinations of mutated sites of those 147 variants is important and beneficial, in the sense that this is a joint consideration among many factors, including computational efficiency, candidate diversity, and the likelihood of discovering improved variants. First, it is infeasible to enumerate and screen all the high-order mutations of TEM-1. The total number of mutants (20^{286} , where 286 is the length of TEM-1 sequence) is beyond what modern computers can enumerate in a reasonable time. We thus resorted to considering a restricted space, i.e., the mutated

site of the 147 “good” variants. To expand the diversity, we consider the re-combinations of those mutated sites (see Methods in our manuscript), which leads to 18,937 variants that are different from our training variants and the 147 variants. As those 147 variants are known to be inhibitor-resistant, we expect that we still have a good probability to discover “good” variants in the small, perturbed region (re-combination of mutations) around them. Admittedly, there are other approaches that do not have to restrict the variation space as did in our work. For instance, the single mutation walk can be iteratively performed to navigate the fitness space and identify better variants (Wu, et al., *PNAS*, 2019; DOI: 10.1073/pnas.1901979116). Our approach here can be viewed as incorporating prior knowledge (the 147 good variants) to help navigate to a good region in the fitness space. Without using the 147 variants to restrict the space, the prioritization could be inefficient and low-yield. We thus think it is a useful step in our approach.

The FoldX-filtering is another quality control step to deliver promising variants. As the stable structure is a prerequisite for a protein to perform meaningful functions, we remove variants that were predicted to be unstable by FoldX and did not synthesize them in the lab.

To summarize, we believe that, in principle, an alternative ENet pipeline without restricting the variation space or performing FoldX-filtering would not limit the quality of variants it can discover. Nevertheless, implementing those two strategies will bring efficiency advantages and offload the experimental burden. Truly validating this requires another round of computational and experimental experiments. We thus leave the ablation study as future work considering our limited resources.

References:

Wu, Zachary, SB Jennifer Kan, Russell D. Lewis, Bruce J. Wittmann, and Frances H. Arnold. "Machine learning-assisted directed protein evolution with combinatorial libraries." *Proceedings of the National Academy of Sciences* 116, no. 18 (2019): 8852-8858.

8. Any justification of the 3 ampicillin concentration levels? How to interpret the various hit rates at various concentration levels? If the hit rates increase with concentration levels, is there any "control" hit rate to be compared to in Figure 5C, so that the reported hit rates of ENet at various concentration levels can be better interpreted?

Response: We thank the reviewer’s comment. We selected the three ampicillin concentrations to cover a low concentration level (300 µg/mL), a high concentration level (1500 µg/mL), and a very high concentration level (3000 µg/mL) to test our mutants’ capacity for ampicillin resistance at different concentration levels of ampicillin. Having these three concentration levels could help us to avoid the potential bias of the mutants’ ampicillin resistance to a specific concentration level of ampicillin. Actually, we noticed that a recent study (Biswas, et al. *Nature Methods*, 2021; DOI: 10.1038/s41592-021-01100-y) also utilized ampicillin concentrations of similar levels (250 µg/mL, 1000 µg/mL, 2500 µg/mL) to ours for the ampicillin resistance assay. Unfortunately, we did not have any "control" hit rate to be compared to in this study. The different hit rate at different ampicillin concentration levels is also the very reason why we selected multiple concentration levels of ampicillin for the assay. We believe this is the most efficient way to reflect the overall landscape of predicted mutants’ resistance capacity to ampicillin.

References:

Biswas, Surojit, Grigory Khimulya, Ethan C. Alley, Kevin M. Esvelt, and George M. Church. "Low-N protein engineering with data-efficient deep learning." *Nature Methods* 18, no. 4 (2021): 389-396.

Minor:

"BARD1 biding" (binding?) appeared twice in Figure 2 and once in Figure S1. "Variants construction missing a "t" in Figure 5a.

Response: Thanks. We have fixed the typos in our revised typos.

Reviewer #2

The authors present a learned model that combines global embeddings of protein sequences with an embedding of the Multiple Sequence Alignment (MSA) of a protein. They apply their model to predict the fitness of variants in a large number of published Deep Mutational Scanning (DMS) datasets, as well as some novel b-lactamase variants, and show that their model achieves marginally better performance than existing methods.

Some similar papers include:

- Unified rational protein engineering with sequence-based deep representation learning, 2019, Alley. Presents the idea that embedded protein representations could be valuable for predicting experimental datasets, and benchmarks them against several DMS datasets.
- Evaluating Protein Transfer Learning with TAPE, 2019, Rao. Applies language models to a variety of prediction tasks, including DMS dataset prediction.
- Low-N protein engineering with data-efficient deep learning, 2021, Biswas. This paper expands upon this idea, and applies these techniques to engineer an evolved GFP and also a TEM-1 b-lactamase

The main contribution of this paper is a novel encoding that combines a global language model representation with an encoding of the protein's MSA. Adding information about homologous sequences of a protein improves the predictive capabilities of the model.

Response: We sincerely thank the reviewer for the summary of our manuscript and the discussion of its position in the context of existing literature.

Overall, I doubt that this paper will have a very large impact on the field. The improvement in performance is marginal, and comes at the cost of restricting the model to work well only for proteins for which excellent MSAs can be obtained.

Response: We first acknowledge that high-quality MSA is beneficial to our ECNet algorithm. As will be illustrated below, ECNet's prediction accuracy positively correlates with the number of sequences in the MSA. Not only ECNet's prediction performance is impacted by the MSA, many existing unsupervised approaches, including EVmutation and DeepSequence that we compared in our manuscript, heavily rely on the statistical signals from a large number of sequences in the MSA to accurately estimate the model parameters. However, in contrast to those methods, ECNet's is not largely affected by the availability of abundant MSA sequences. Due to the supervised nature of ECNet, the weak signals of an MSA with a low number of sequences can be mitigated by the direct signals from DMS data for function prediction: in addition to using the MSA evolution contexts as features, ECNet also uses the DMS fitness data as supervision signal to train the ML model. Therefore, the model does not have to solely rely on the MSA data but can utilize the DMS data to predict the fitness. As will be shown in the response to question Q2 below, ECNet is still able to predict fitness accurately (>0.8 correlation) even though that protein only has very few homologous sequences (e.g., <10k sequences). As a result, we think that ECNet would not be severely restricted to proteins for which excellent MSAs can be obtained.

The main result - that adding in MSA information improves predictive capabilities - is not a surprise: the highly related field of machine learning-based protein structure prediction has observed this numerous times.

Response: We thank the reviewer for reviewing the protein structure prediction field in which MSA data is widely used to inform the machine learning model (e.g., AlphaFold 2). However, integrating evolutionary information to predict protein fitness is relatively new in the field of ML-assisted protein engineering. Our work, to our best knowledge, for the first time demonstrates that global and local evolutionary contexts can be integrated to facilitate supervised deep learning models for protein fitness prediction. After the initial submission of our manuscript, there are at least two new studies that share similar ideas as our work: Hsu et al. (2021) developed a model that combines local evolutionary data with (semi-)supervised models, and Frisby et al. (2021) proposed an algorithm to combine global evolutionary contexts with supervised models in a sequential manner. Both studies validated that adding evolutionary information is beneficial for function prediction in protein engineering, and our work differs from them in that we integrated *both* global and local evolutionary contexts with an LSTM-based deep learning model.

References:

Hsu, Chloe, Hunter Nisonoff, Clara Fannjiang, and Jennifer Listgarten. "Combining evolutionary and assay-labelled data for protein fitness prediction." *bioRxiv* (2021).

Frisby, Trevor S., and Christopher James Langmead. "Bayesian optimization with evolutionary and structure-based regularization for directed protein evolution." *Algorithms for Molecular Biology* 16, no. 1 (2021): 1-15.

The conceptual contribution of proposing that protein embeddings can be used to predict experimental datasets has been made already in numerous papers, and there have already been many iterations on exactly how the embedding should be constructed (several of which they cite, such as TAPE).

Response: We agree with the reviewer that learning protein representations (i.e., global evolutionary contexts used in our method) has been an active direction of protein engineering and machine learning. In our work, we demonstrated that those global evolutionary contexts (e.g., TAPE or esm-1b) can be combined with some more specific information (i.e., local evolutionary contexts learned from MSA) to better inform the deep learning model, which distinguishes our work from those that only focus on learning global embeddings.

This paper even uses the same protein evolution test case as a previous paper: b-lactamase evolution.

Response: We chose beta-lactamase for the following reasons: 1) it has a good number of screened variants in the literature that we can use as the initial training data to build our model; 2) it is a suitable test case such that we can efficiently construct the variants and screen their fitness in our wet-lab environment. While it is not a completely new test case that has not been engineered in previous papers, we think beta-lactamase is a good proof-of-concept to demonstrate the utility of our deep learning model in the actual experimental setting.

Overall, it is a nicely put together manuscript, but does not offer very strong conceptual or methodological advances.

Response: We are glad that the reviewer thought positively about our manuscript. As our manuscript for the first time demonstrated how global and local evolutionary contexts can be integrated into supervised deep learning models and has achieved promising results both computationally and experimentally, we believe it will be a contribution to the literature and be of interest to readers in the broader scientific field.

Some things that would strengthen the paper:

1. It would be helpful to do analysis on your own model to confirm that the improved performance is because of the MSA. There's a section on ablation analysis in figure S2, could you expand upon this somewhat? From this graph, it seems that the local evolution representation alone performs on-par with ECNet. It is unclear to me whether this is the analysis I'm looking for.

Response: We thank the reviewer's comment. In **Figure S2** (now labeled as **Figure S3** in our revised manuscript), we did ablation analysis to compare ECNet to other models that only used global evolutionary features or local evolutionary features. We replotted the data in **Figure R6a** as a scatter plot to make the comparisons more explicit. In this scatter plot, we compared the full ECNet model with two variants, one dropped the local evolutionary features and the other dropped the global evolutionary features. In both cases, the full model significantly outperformed the individual model (one-sided rank-sum test $P > 10^{-6}$). This suggested that both the global and local evolutionary features contributed to the performance improvements. In particular, we found that the model that did not use the local evolutionary features has more correlation degrades compared to the model without global evolutionary features (mean $\Delta\rho=0.081$ v.s. $\Delta\rho=0.042$), which suggested that the local evolutionary contexts provided more additional values to the prediction improvements than the global evolutionary contexts. This is likely because the local evolutionary features extracted from MSA contained more specific features of the target protein while the global evolutionary features only learned general features for protein sequences in Pfam and are less specific.

The analysis above was conducted on the DMS data of 39 proteins in the DeepSequence dataset, most of which are single-mutation variant data. We then repeated the ablation analysis on several double-mutation datasets. We had a similar observation on those double-mutation data (**Figure R6b**). That is, using the global or local evolutionary features only did not match the prediction performance to a model that used both, and the local evolutionary features contributed more to the improvements than the global evolutionary features.

To summarize, our ablation results suggested that both the global and local evolutionary features improved the prediction performance on most of the proteins. The local evolutionary features contributed more to the improvements than the global features, partially because they captured signals that are more specific to the interested protein (e.g., amino acid preferences at individual sites and pairwise co-evolution relationships).

2. How does performance depend on the number of sequences in the MSA and the quality of the MSA? You have some data about this in figure 1b, but it would be nice to see more nuanced analysis.

Response: We thank the reviewer's comments. As the same question has also been raised by Reviewer 1 (Question 2), here we re-used some of the figures and texts in our response to that question. We have performed several new analyses to investigate how ECNet's prediction performance depends on the data it used, including the MSA and DMS (deep mutational scanning) data. In short, we found that (i) at the protein level, we found that ECNet's prediction performance is better for proteins with more homologous sequences; (ii) at the residue level within a protein, we found that ECNet predicts fitness better for sites that are more conserved; and (iii) by comparing ECNet with another existing method, we additionally found that ECNet gained more improvements if the training DMS dataset is more complete. We provide our detailed results below.

Prediction performance correlates with the number of MSA sequences. First, we expanded the results in **Figure 1b**, where we used the residue co-variation derived from MSA to predict the fitness of single-mutation variants of proteins in the Envision dataset. Recall that in this experiment, a direct coupling analysis model was used to fit the MSA data and the change of the co-variation terms upon

mutations was used to predict the fitness data (i.e., in an unsupervised manner). Here, we replotted the results in a scatter plot in **Figure R7**. It is clear that the number of MSA sequences correlated very well with the prediction performance (Spearman's $\rho=0.82$) because more sequences in the MSA gave more accurate and robust estimations about the amino acid preference at each site in the sequence.

Next, we extended the analysis to the benchmark results on the DeepSequence dataset, which consists of DMS data of 38 proteins. We found that ECNet's prediction performance still positively correlated with the number of sequences in the MSA (**Figure R8a**; Spearman's $\rho=0.27$). The correlation coefficient was lower than our analysis above for the Envision dataset in **Figure R7**. One major reason is that, unlike the unsupervised approach in **Figure R7** that entirely relied on MSA to predict fitness, here our ECNet algorithm is a supervised model and utilized both MSA data and training fitness data to predict fitness. Therefore, for some proteins, ECNet is still able to predict fitness rather accurately even though that protein only has very few homologous sequences in MSA. For example, ECNet achieved a correlation >0.8 for several proteins with less than 10k MSA sequences (**Figure R8a**).

Less-conserved sites are easier to predict. Motivated by the reviewer's comments, we further investigated what other factors might impact ECNet's performance. Proteins do not mutate in a uniform way in the evolution, i.e., some sites in the sequence are more conserved towards specific amino acids

(AAs) while others can tolerate different AAs. Therefore, we hypothesized that the mutation effects at different sites are not equally ‘hard’ to predict. We thus plotted the relationship between the entropy at a site in the sequence and ECNet’s prediction accuracy for that site (**Figure R8b**). The entropy value is calculated using the AA frequency at each site (column) in the MSA and quantifies the conserveness of each site -- a lower entropy means this site has a dominant preference towards specific AAs while a higher entropy indicates that many AAs are likely to appear at this position (in terms of occurrence frequency). We observed that conserved sites (lower entropies) are easier to predict at higher correlations (**Figure R8b**; $\rho=-0.34$). This is because, for conserved sites, the mutation (no matter what AA it mutates to) is likely to be damaging, which is more straightforward for the model to predict. In contrast, for tolerable sites, the mutation consequence can be damaging, neutral, or even beneficial, raising more ambiguity for the predictive model.

the completeness of single-mutation protein DMS data in the DeepSequence dataset. The completeness is defined as the percentage of screened variants among all possible single-mutation variants for a protein, i.e., $M/(19*L)$, where M is the number of screened variants in the DMS data and L is the length of the protein sequence. Each point represents one protein in the DeepSequence dataset. ECNet's prediction performance is summarized by the average of five-fold cross-validation. **(d)** Same as **(c)** except that the DMS completeness is defined for each site. The per-site completeness is defined as the percentage of screened mutations among all possible single-mutation mutations for a position, i.e., $m/19$, where m is the number of screened mutations for this position in the DMS data. Each point represents one protein in the DeepSequence dataset. The per-site completeness is first averaged over all test sites in a fold and then averaged over all five folds. (MSA: multiple sequence alignment; DMS: deep mutational scanning.) This figure is labeled as **Figure S9** in the revised manuscript.

DMS data completeness influences model accuracy. Finally, we analyzed for which proteins has ECNet gained more improvements compared to DeepSequence, an existing unsupervised method. As ECNet is a supervised algorithm, we hypothesized that its prediction accuracy is impacted by the quality of training data. Here, we analyzed the relationship between the model's prediction performance and the completeness of a DMS dataset. A protein with sequence length L has $19*L$ possible single-mutation variants. However, in practice, a DMS study may not always be able to sample all the possible single-mutation variants due to technical reasons. Using the single-mutation DMS data of 37 proteins from the DeepSequence dataset, we plotted the completeness of the DMS data and ECNet's prediction performance in a scatter plot (**Figure R8c-d**). For both sequence-level and per-site completeness (see **Figure R8** legend for definitions), we observed that ECNet trained on more complete DMS data has achieved better results. As a more complete DMS dataset samples more variants, either covering more sites or screening more mutations on the same site, the ECNet models trained on this dataset are thus more accurate for a wide range of sites and mutations. A recent concurrent work also obtained a similar observation (Wittmann et al, *bioRxiv*, 2020; DOI:10.1101/2020.12.04.408955).

To conclude, our new analyses here dissected the factors that impact ECNet's prediction performance. We found that both MSA characteristic and DMS data properties can impact the prediction accuracy. We added the result figures to the Supplementary Information and the corresponding text to the Discussion section in our manuscript, which is also reproduced in the box below.

Revised text (Section "Discussion"):

ECNet's prediction performance is impacted by the MSA characteristics and DMS data properties. For example, ECNet predicts better for proteins with more homologous sequences (**Figure S8** and **Figure S9a**), for sites that are more conserved within a protein (**Figure S9b**), and for proteins that have a more complete DMS dataset (**Figure S9c-d**).

References:

Wittmann, Bruce J., Yisong Yue, and Frances H. Arnold. "Machine learning-assisted directed evolution navigates a combinatorial epistatic fitness landscape with minimal screening burden." *bioRxiv*, (2020).

3. How do you determine the number of parameters in the top model? Do you adjust the number of parameters in response to the size of the experimental dataset?

Response: We performed an inner-loop grid search using the training data of a target protein to decide the hyperparameters of ECNet. No test data was used to decide the hyperparameters. While the optimal hyperparameters varied slightly for individual proteins, we found that there was a set of reasonably good default values that can be used as a starting point for a new protein. Nevertheless, careful grid search of hyperparameters would further improve the model performance. We have revised our manuscript as below to include this information.

Revised text (Section “Methods - Training details”):

To select the hyperparameters of ECNet, we performed a small-scale grid search using the training data of a protein, such that $\frac{7}{8}$ of the training data was used to train a model with a specific set of hyperparameters and the remaining $\frac{1}{8}$ data was used as the validation set to select the hyperparameters. The test set was not used for hyperparameter selection. In particular, we tested the LSTM’s dimension of $d_{LSTM} = 32, 64, \text{ and } 128$, the top layer dimension of $d_h = 32, 64, \text{ and } 128$, the reprojected embedding dimension of $d_{proj} = 128 \text{ and } 256$. In general, we found that $d_{LSTM} = 128$, $d_h = 128$, and $d_{proj} = 128$ are reasonably good defaults and can be used for a new protein. Nevertheless, careful grid search of hyperparameters for the new protein would further improve the model performance. Unless otherwise specified, the batch size was set to 128 and the maximum number of training epochs was set to 2,000 with an early stop if the performance has not been improved for 1000 epochs. Model training was performed on an Nvidia TITAN X GPU.

4. How does this model respond to noisy experimental data? If you add x% noise to the training data on one of the datasets, how does it impact the spearman correlation? We might expect that some models are more brittle/robust to experimental data noise - this would be a valuable thing to measure and I'd be interested to see if your model out-performs others.

Response: We thank the reviewer for the great suggestion on evaluating the model’s robustness against data noise. In our revised manuscript, we have added a new experiment to simulate noises in the training data and evaluated ECNet’s performance when trained on noisy data. We found that ECNet was robust across multiple noise levels (for example, ECNet’s performance only decreased by 2% when the training data was perturbed by 10%). In addition, ECNet still outperformed other methods when trained on noisy sets. We provide our results and analyses in detail below.

ECNet is robust against noise in training data. We simulated experiments where ECNet was trained on noisy training data and tested on noise-free data. Specifically, we perturbed the training data by adding Gaussian noises to the fitness data as follows: $\hat{f}_i = \alpha_i f_i$, where α_i is sampled from the normal

distribution $N(1, \sigma)$. The perturbation level was controlled by the standard deviation σ of the Gaussian noises. We simulated three training sets by varying σ at 0.1, 0.25, and 0.5 (see **Figure R9** and the figure legend for details). To quantify the perturbation effect on the fitness data of each protein DMS dataset, we define a “data inconsistency” score to indicate how far the perturbed fitness data deviated from the original data. The inconsistency score is defined as $1 - r$, where r is Spearman’s correlation between the pre- and post-perturbation fitness data. For example, for noise level $\sigma=0.5$, the inconsistency scores of the perturbed 39 DMS datasets in the DeepSequence dataset ranged from 0.05 to 0.76, with a mean of 0.20 (**Figure R9b** top), which was a significant perturbation. We trained an ECNet model on the perturbed training data for each protein and evaluated its performance. We found that, compared to the ECNet model trained on unperturbed data (**Figure 3** in the main text), the ECNet model trained on noisy data ($\sigma=0.5$) suffered a correlation drop ranging from 0.05 to 0.24, with a mean drop of only 0.07 (**Figure R9b** bottom). This result suggested that ECNet is relatively robust against the noise in training data -- its absolute performance decrease was only 7% when the training data was perturbed significantly by 20%. In **Figure R9c**, we showed the relationships between training data inconsistency and test correlation drop for all 39 proteins in the DeepSequence dataset. For noise levels $\sigma=0.1, 0.25, \text{ and } 0.5$, the training data inconsistency was 0.03, 0.09, and 0.20, respectively, while ECNet’s performance drop ($\Delta\rho$) was 0.01, 0.02, and 0.07, respectively. This result suggested that ECNet is very robust against different levels of noise, with a performance drop as negligible as 1%.

ECNet is more robust than other methods. We compared ECNet to the TAPE and One-hot models at noise levels $\sigma=0.1, 0.25, \text{ and } 0.5$ (**Figure R9d**). We first noticed that the One-hot model’s test correlation rapidly decreased when noises were injected into the fitness data, which implies that the sparse one-hot representations are easy to overfit to the fitness data and sensitive to data perturbations. In contrast, ECNet’s performance remained relatively stable as the noise level σ increased. The primary reason is that ECNet integrated local and global representations that encode rich features such as evolutionary, structural, and biophysical properties. The information-rich features alleviate the overfitting issue and make the deep learning model less sensitive to data noise. Furthermore, we found that, across all noise levels, ECNet also outperformed another baseline model TAPE, which only uses the global representations for function prediction.

Overall, our experiments above suggested that ECNet is not sensitive to noise in the training data and its prediction performance is more robust than other methods. We have added the above results into our revised manuscript.

Revised text (Section “Results - Accurate prediction of functional fitness landscape of proteins”):

Furthermore, we simulated experiments where ECNet was trained on noisy training data and tested on noise-free data. We found that ECNet was robust against data noise. For example, ECNet’s test correlation only decreased by 2% when the training data was perturbed by 10% (**Figure S4**).

using $\sigma=0.1$, 0.25, and 0.5, and a separate ECNet model is trained on each of the training sets separately. The trained models are then evaluated on the noise-free test set. **(b)** Evaluation results for 39 proteins in the DeepSequence dataset for $\sigma=0.5$. Top: data inconsistency caused by the perturbation for each protein. The inconsistency is defined as the $1 - r$, where r is the Spearman's correlation between the pre- and post-perturbation training set. A larger inconsistency score means the perturbed fitness data have deviated more from the original fitness values. Bottom: ECNet's performance drop when trained on the noisy training data as compared to trained on noise-free training data. The performance is evaluated using Spearman's correlation. A smaller absolute value ($\Delta\rho$) means the model is more robust to errors in the training data. **(c)** A scatter plot that shows the relationship between the training data inconsistency and performance drop for every protein in the DeepSequence dataset, at noise levels $\sigma=0.1$, 0.25, and 0.5. **(d)** A boxplot that compares the prediction performance of ECNet, TAPE, and One-hot models on the DeepSequence dataset at noise levels $\sigma=0$, 0.1, 0.25, and 0.5. The noise level $\sigma=0$ means the model is trained on the noise-free training set. This figure is labeled as **Figure S4** in the revised manuscript.

5. In Figure S3, it would be great to add the performance of a simple one-hot encoded model.

Response: Thanks for the reviewer's suggestion. We have added the performance of a One-hot model into **Figure S3** (now labeled as **Figure S5** in our revised manuscript) in our revised manuscript, which is also reproduced below in **Figure R10**. We found that the One-hot model's performance is substantially outperformed by ECNet and UniRep, and only on par with EVmutation. These results suggested the advantage of biologically inspired representations learned in ECNet and UniRep. In contrast, the sparse, biology-agnostic one-hot representations tended to overfit the training data and have poor generalization ability.

Figure R10. Simulation results of using ECNet to prioritize high-performing variants for avGFP, GB1, and Pab1. (a) Recall versus sequence testing budget curves for each method. The recall is defined as the fraction of the true top-100 variants identified by each method's ranking of variants with the given budget. ECNet was compared to i) UniRep, a supervised method, ii) One-hot, a supervised method that uses simple one-hot sequence representations, iii) EVmutation, an unsupervised method, iv) random model, which is a null model that assigns a random ranking to test variants, and v) ideal model, which ranks the variants using the ground-truth fitness score. (b) Maximum normalized fitness rank observed versus sequence testing budget curves for each method. Fitness scores of variants were normalized based on their rank into a value between 0 (the lowest fitness score) and 1 (the highest fitness score). (c) Efficiency gain of ECNet, UniRep, One-hot, and EVmutation over the random model

with the given testing budget. The efficiency gain is defined as the ratio of a method's recall divided by the recall of the null model as a function of the testing budget. Error bands in **(a)** and **(b)** depict ± 1 standard deviation calculated over 10 independent replicates of the experiments. Curves in (c) were smoothed using an averaging window of size 50. This figure is labeled as **Figure S5** in the revised manuscript.

Reviewer #3

In this work, the authors developed one machine learning-based evolutionary context-integrated neural network (ECNet) technique to exploit evolutionary contexts to predict functional fitness in protein engineering. The algorithm integrates the evolutionary information from the homologous sequences and the evolutionary context from the enormous protein sequence universe to map the sequence to the function of the protein mutants. The authors claim the superior performance of ECNet compared to existing machine learning algorithms by using ~50 deep mutagenesis scanning and random mutagenesis datasets. Also, the authors validated their method experimentally on identifying the variants of TEM-1 β -lactamase through the engineering process.

Response: We thank the reviewer for a nice summary of our work.

While the manuscript is well presented but few portions need more clarification.

1. The strength of pairwise residue dependencies does not correlate well with the fitness of double mutants (Figure 1A). Still, authors rely on this correlation to build their method. A justification for using this low correlation is required.

Response: While ECNet is built on the pairwise residue dependencies, its performance is not upper-bounded by this correlation, because ECNet does not rely on the residue dependencies to predict the final output. The residue dependencies are only used as the input representation of protein sequence to the neural network. To predict protein function, ECNet uses DMS (deep mutational scanning) data that measures the function of interest as the direct supervision signal to train the neural network. Therefore, our model's performance is not bounded by the correlation achieved by the pairwise residue dependencies.

Despite the residue dependency's correlation seems low (Spearman's ρ -value 0.35) in **Figure 1a**, using them as representations of protein sequences provides additional and orthogonal values for the machine learning model (see ablation studies in **Figure S3** in our revised manuscript). We also noted that the 0.35 correlation was only for a particular protein (WW domain) rather than general proteins, and a value of 0.35 was not regarded as a low correlation for unsupervised protein function prediction in a previous study (Riesselman et al., *Nature Methods*, 2018; DOI: 10.1038/s41592-018-0138-4). Given that the dependency information had already achieved a non-trivial correlation of 0.35 when used in an unsupervised manner, we hypothesized that it encodes features useful for predicting protein function and the correlation will be further improved when the dependency information is integrated into a supervised machine learning model. And that is exactly what ECNet did. For example, for the same protein in **Figure 1a**, we boosted the correlation to 0.72 from 0.35 after we integrated the dependency information into a supervised model (**Figure 4a**) and outperformed another supervised model that did not use the dependency information (correlation 0.61).

In summary, the non-trivial correlation achieved by the pairwise residue dependencies does not limit the performance of our model, rather, it encodes useful information for function prediction. When used as

prior knowledge (in the form of protein sequence representation), this information better informs the supervised learning model and improves the prediction performance.

References:

Riesselman, Adam J., John B. Ingraham, and Debora S. Marks. "Deep generative models of genetic variation capture the effects of mutations." *Nature methods* 15, no. 10 (2018): 816-822.

2. Despite the dataset section, it is not clear what is the training data, what is the testing data, what fraction of the data is a positive sample, what fraction is a negative sample. Also, it is not clear whether the dataset is balanced or not. Since any ML-based or DL-based technique heavily depends on the dataset, so more details are required for the dataset.

Response: We listed our train/test splitting of each dataset/experiment below. We also included those details in our manuscript.

- *DeepSequence* (**Figure 3**): This dataset was used as the major benchmarking experiment to assess our model's prediction performance and to compare it to other leading methods. Therefore, we performed five-fold cross-validation on this dataset, where the fitness data of a protein was randomly partitioned equally to five folds (20% of the data) and in each round four folds were used as the training data and the remaining fold was used as the set. The average prediction performance over the five folds of each protein was reported.
- *Envision* (**Figure 2**). This dataset was used to compare our representation learning performance to other representation approaches. The data was collected from a previous study (Gray et al., *Cell Systems*, (2018)). We thus followed the data splitting strategy there, i.e., 20% of the data was randomly reserved for testing and the remaining 80% was used for model training.
- *Variant prioritization* (**Figure S5**). This experiment was designed to assess how accurate and efficient ECNet is in retrieving high-performing variants. We thus collected three large deep mutation scanning datasets (>10k variants) of three proteins. For the data of each protein, 90% was randomly sampled as the training data and the remaining 10% was used as the test set.
- *Viral protein dataset* (**Figure S6**). This dataset was used to evaluate the unsupervised version of ECNet. Therefore, all quantitative fitness data in the dataset was used as the test set, and no data was used directly in model training.
- *Effects of training data size* (**Figure S7**). This experiment was designed to study the effect of training data size on model performance. We first withheld 10% (randomly sampled) data as the test set. For the remaining 90% data, we trained two ECNet models, where one used all of them as training data and the other only used ¼ of them as training data.
- *Model generalizability* (**Figure S10**). To evaluate ECNet's generalizability, we used a site-wise and a AA-wise split strategy to create train/test sets. This represents a more challenging evaluation setting than the random split. For example, in the site-wise split strategy, the model will be evaluated on sequence positions that it has not been trained on to test its generalizability to variants on unseen sites. More details can be found in the legend of **Figure S10**.

Regarding the fraction of positive/negative samples, we would like to note that for all of our analyses/experiments except the one in **Figure 2b**, we cast the prediction task as a regression problem, so we did not introduce a notion of positive and negative samples in those experiments. We train the model to predict the fitness value of every variant, instead of differentiating positive from negative variants. For the experiment in **Figure 2b**, we cast the prediction task as a classification problem, so we introduced a notion of positive/negative for each variant and the model is trained to classify a variant to positive or negative samples. A variant is said to be positive if it has an improved fitness value than the wild type and negative otherwise (see more details of the definition below in our response to comment Q7). The fraction of positive samples in the data of each protein is shown below in **Figure R11**.

3. The authors mentioned utilizing different datasets from different published works. However, it is not clear in total how much data is utilizing for ECNet. There is a possibility that the dataset may have some similarities. Did the author check for data duplicates or data redundancy? An explanation is required in this context.

Response: The datasets (a dataset is defined as the DMS data of a particular protein) from different publications were used independently/separately for evaluation in our work. We have performed two types of evaluation experiments: (i) cross-validation using a single dataset (deep mutagenesis scanning data of a particular protein), and (ii) training the model on a single-mutation dataset and testing it on a double-mutation dataset, with both datasets measuring the fitness of the same function. For both types of experiments, we ensured that there are no data duplicates or data leakage between the training and test sets. Specifically, in the type (i) evaluation experiment, we split the data into non-overlapping

training/test sets and performed cross-validation *within* each dataset (i.e., the data of a single mutagenesis study for a particular protein), and no data from other proteins was used to train the model. In type (ii) experiment, only single-mutation fitness data was used to train the model and double-mutation fitness was used to test the model, which means there is no data duplicates/redundancy across the training and test sets. To conclude, we ensured that there is no data leakage or redundancy between the training and test sets. We have clarified this in our revised manuscript.

4. The authors have used a language model for protein sequence. Despite the results, an analysis (or explanation) is required regarding the validation of such use since it is the core part of their ML-based method. The authors may include this in the introduction part.

Response: The language model is used to capture the underlying constraints of protein sequence, just like the grammar or semantics existing in human languages. Over the course of evolution, nature samples protein sequences that it deems as “useful” proteins, for example, those that preserve or improve the protein's fitness, such as stability, structure, and function. The underlying distribution of protein sequences is thus shaped by evolution and reflected by the natural protein sequences we observed today. We used a language model (LM) to capture this underlying distribution. This is inspired by the great success of LMs in natural language processing, e.g., BERT (Devlin et al., *arXiv preprint arXiv:1810.04805* (2018)) and GPT-3 (Brown et al., *arXiv preprint arXiv:2005.14165* (2020)), which are capable of learning the complex rules and meanings of human languages. Similarly, we used LMs for proteins and to unravel the ‘grammars’ or ‘semantics’ of sequence generation by evolution. LMs have been applied to model protein sequences and to predict the effects of mutations in several studies in the past two years, e.g., predicting viral escape (Hie, et al. *Science*, 2021; DOI: 10.1126/science.abd7331), protein structure and function (Rives et al., *PNAS*, 2021; DOI: 10.1073/pnas.2016239118).

We revised the introduction part of our manuscript to include the above explanation, as shown below:

Revised text (Section “Introduction”):

More recently, inspired by the advances in natural language processing, a trend is emerging that pre-trains a language model (LM) on large protein sequence datasets to learn the distribution of protein sequences^{13,16–22}. The protein sequences observed in nature today are the results of natural selection by evolution. Out of the possible mutations to a sequence, evolution samples those that preserve or improve the protein's fitness, such as stability, structure, and function. The underlying constraints or factors that determine protein's fitness have shaped the distribution of protein sequences. LMs are used to unravel the ‘grammars’ or ‘semantics’ of sequence generation by evolution. By being trained on natural sequences to predict the likelihood that a particular amino acid appears within a context, the language model learns representations that are semantically rich and encode structure, evolutionary and biophysical contexts¹⁶. It was found that using the learned

representation as the feature input to fine-tune a supervised model improves fitness prediction on multiple protein mutagenesis datasets¹⁷.

References:

Devlin, Jacob, Ming-Wei Chang, Kenton Lee, and Kristina Toutanova. "Bert: Pre-training of deep bidirectional transformers for language understanding." *arXiv preprint arXiv:1810.04805* (2018).

Brown, Tom B., Benjamin Mann, Nick Ryder, Melanie Subbiah, Jared Kaplan, Prafulla Dhariwal, Arvind Neelakantan et al. "Language models are few-shot learners." *arXiv preprint arXiv:2005.14165* (2020).

Hie, Brian, Ellen D. Zhong, Bonnie Berger, and Bryan Bryson. "Learning the language of viral evolution and escape." *Science* 371, no. 6526 (2021): 284-288.

Rives, Alexander, Joshua Meier, Tom Sercu, Siddharth Goyal, Zeming Lin, Jason Liu, Demi Guo et al. "Biological structure and function emerge from scaling unsupervised learning to 250 million protein sequences." *Proceedings of the National Academy of Sciences* 118, no. 15 (2021).

5. Authors mentioned that "Using a large corpus of protein sequences such as UniProt and Pfam, a language model learns to predict the next amino acid given all its preceding amino acids." Again a justification is required since one or a few critical mutations may destroy the protein structure.

Response: A language model (LM) predicts the likelihood that a particular amino acid appears at a position given the surrounding context. If a mutation is very critical and can destroy the protein structure, the evolution will not select this mutation as it is not beneficial for the evolution itself. As a result, we will rarely see this mutation in the protein sequence existing in nature today. The LM can learn to capture the effect of a critical mutation from data and predict a very low probability for that specific mutation. Therefore, through the representation learned by LM, our algorithm ECNet can exploit this mutation effect to predict protein function.

It should be noted that the mutation effects encoded in the LM are general properties for all proteins in the training data, i.e., the protein sequences in UniProt or Pfam. As the LM is trained on all sequences in the sequence corpus, the mutation effects captured by the LM are broadly applicable to a wide range of proteins. However, there can be mutation effects that are specific to a particular protein (or a protein family), i.e., one mutation is critical for this protein only but is tolerable in other proteins. The LMs may not be very sensitive to such effects specific to a protein, as the statistical signals are weak and may be "averaged out" by a large number of sequences of other proteins. Nevertheless, in this case, ECNet still can exploit the other side information, i.e., fitness quantified by deep mutational scanning (DMS), to learn the mutation effects.

In summary, LM can capture the effect of critical mutations and inform ECNet's learning process. If the mutation effect is too specific to a protein and not captured by the LM trained on general protein sequences, ECNet can rely on DMS data as supervised signals to learn the mutation effects and to predict the function.

6. The authors evaluated the performance of their model using 5-fold cross-validation (Figure 2). How many random experiments were performed? It would be good to iterate it for 100 runs and present the mean and standard deviation of the runs. Apart from relative improvement (Figure 2b), please provide the absolute values of AUROC.

Response: We thank the reviewer’s suggestion. We repeated the cross-validation by 10 times and presented the mean AUROC values and standard deviations (S.D.) in **Figure R12** below (**Figure 2b** in our revised manuscript). We did not repeat the experiments up to 100 runs because i) it would take weeks given our current computational resources and ii) we think 10 repetitions of five-fold cross-validation have already been an accurate estimation of the performance. Similarly, we also presented the mean and S.D. of relative improvement of Spearman correlation in **Figure S2** in our revised manuscript. The absolute values of AUROC and Spearman correlation were provided in **Table R1 (Supplementary Table S3)** in the revised manuscript).

Protein	AUROC		Spearman’s correlation	
	ECNet	Envision	ECNet	Envision
BRCA1 (E3 activity)	0.32	0.29	0.41	0.33

BRCA1 (BARD1 binding)	0.54	0.52	0.43	0.38
Ubiquitin (E1)	0.58	0.53	0.73	0.71
UBE4B (U-box)	0.49	0.41	0.56	0.48
PSD95 (pdz3)	0.86	0.82	0.80	0.75
Pab1	0.45	0.36	0.81	0.80
TEM-1	0.59	0.55	0.86	0.87
Ubiquitin	0.73	0.65	0.87	0.82
Yap65 (WW)	0.79	0.74	0.75	0.72
Protein G	0.87	0.83	0.91	0.87
HSP90	0.55	0.52	0.70	0.67
KKA2	0.80	0.88	0.80	0.88

Table R1. Raw performance numbers on the Envision dataset. Performances of ECNet and the Envision model are evaluated using AUROC and Spearman’s correlation. The mean values of ten trials of five-fold cross-validation are listed. This table is labeled as **Supplementary Table S3** in the revised manuscript.

7. “As protein engineering focuses on identifying variants with improved properties than the wild type, we further evaluated the model performance using a classification metric (AUROC score), in which variants with higher function measurements than the wild-type sequence are defined as positive samples and the remaining variants as negative samples.” How relevant is it in differentiating positive and negative samples? What is the threshold value used?

Response: This evaluation experiment was designed to directly assess the model’s ability to differentiate positive from negative samples. The prediction task was cast as a classification problem, where the model is asked to predict whether a variant is positive or negative. The AUROC score was used as the metric to assess the prediction performance, and a score of 1.0 means perfect prediction and a score of 0.5 represents a random guess.

A variant is defined as positive if it has a better fitness value than the wild type, otherwise, it is called a negative sample. In this evaluation experiment, we used the Envision dataset curated in a previous study (Gray et al., *Cell Systems*, 2018; DOI: 10.1016/j.cels.2017.11.003), where the fitness values of protein variants have been normalized such that the most damaging variants have a fitness ~ 0 while the most wild-type-like variants have a fitness ~ 1.0 . We thus labeled variants with ≥ 1.0 fitness scores as positive samples and the remaining ones as negative samples. The model’s predictions were compared against those binary labels to compute the AUROC scores.

References:

Gray, Vanessa E., Ronald J. Hause, Jens Luebeck, Jay Shendure, and Douglas M. Fowler. "Quantitative missense variant effect prediction using large-scale mutagenesis data." *Cell systems* 6, no. 1 (2018): 116-124.

8. While mapping the sequence to function, whether authors consider the stability of the protein? How do they are certain about stability? Specifically, when authors comment that “to assist protein engineering, the machine learning model needs to prioritize variants that are not only structurally stable and non-deleterious but also with enhanced properties.”

Response: In this work, we included a biophysical computational model (FoldX), as a quality-control step of ECNet, to find variants that are structurally stable. Specifically, in the case study of engineering new TEM-1 variants with ECNet, we first used ECNet to predict the fitness for a set of candidate variants and then applied FoldX to predict the change of stability ($\Delta\Delta G$) upon mutations in each variant. Next, we selected variants that have both a high fitness value (predicted by ECNet) and a stable structure (small $\Delta\Delta G$, predicted by FoldX). The selected variants were then sent to experimental validation.

9. In the dataset section, authors are requested to differentiate training set from the testing set and positive data from negative data - specifically for viral proteins.

Response: For training/test set splitting, we have provided the details in our response to question Q2 above and revised the manuscript accordingly.

Regarding differentiating positive data from negative data, we would like to clarify that for all of our analyses/experiments except the one in **Figure 2b**, we cast the prediction task as a regression problem. So we did not introduce a notion of positive and negative samples in those experiments. The prediction results were evaluated using Spearman correlation, a widely used metric for a regression problem. For the experiment in **Figure 2b**, we cast the prediction task as a classification problem, so here the variants should be defined as positive or negative samples. To this end, we defined a variant with a fitness value higher than that of the wild type as a positive sample, otherwise, as a negative sample. Note that the data used in this experiment has been normalized such that the variants with the most disruptive mutations have a fitness value of 0, and the wild-type-like variants have a fitness of 1.0 (see our response to comment Q7 above). We thus used 1.0 as the threshold to differentiate variants into positive and negative samples.

Specifically, for viral proteins, we again cast the prediction task there as a regression problem, and we encouraged the model to accurately predict the fitness value for every variant, instead of classifying “positive” variants from “negative” ones. Therefore, we did not have a notion of positive/negative samples for viral proteins in that experiment (**Figure S6**).

10. On page 21, what is the difference between N and L? Does $L + (\text{\#gaps}) = N$? Please add few lines for describing L more clearly.

Response: L is the length of our target protein sequence. We used this protein sequence as the query to search for homologous sequences. The number of those homologous sequences is denoted as N. In other words, if we think of the MSA as a matrix with the shape “number of sequences by number of amino acids”, N and L are its two dimensions, respectively. Whether L includes gaps depends on the actual MSA format. In this work, we constructed MSA using the A3M alignment format, in which the query sequence does not contain gaps. So, here, L is exactly the number of amino acids of the wildtype sequence of our interested protein. We have clarified this in the revised manuscript.

11. How do you achieve $O(NL^2)$? Few lines on the derivation will be good.

Response: The $O(NL^2)$ optimization is achieved by applying a technique called pseudo-likelihood maximization (PLM) to infer the Markov random field model (also called Potts model, or Ising model). PLM has been extensively used for protein structure/contact prediction previously (Kamisetty et al. *PNAS* 110.39 (2013); Ekeberg et al. *Phys. Rev. E*, 2013, vol. 87). The key idea is to approximate the likelihood, which takes exponential time to compute the partition function, by a pseudo-likelihood, which is the conditional probability of a variable given other variables and only takes polynomial time to compute the partition function. Specifically, the optimization iterates over all sequences in the MSA ($O(N)$ time); and for each position in the sequence ($O(L)$ time), it computes the conditional probability of an amino acid showing at this position, given all amino acids at other positions fixed. It takes $O(L)$ time to compute this partition function of this probability. So the overall running time is $O(N*L*L)$. In the manuscript, we also referred our readers to those papers for the details of the optimization.

12. In the dimensionality reduction step, how do you decide on d? What is the value for d? Is it protein sequence invariant?

Response: The value of d is determined by performing a grid search over $d=\{32, 64, 128\}$ using inner split on the training data, such that $\frac{7}{8}$ of the training data was used to train a model with a specific value of d and the remaining $\frac{1}{8}$ data was used to evaluate the performance and decide the choice of d. The test data is not used in this process. We have the “Training details” section in our revised manuscript to reflect this detail. More information about the choice of hyperparameters is provided in our response to Q16 below.

13. Please draw the architecture diagram for Sequence-to-function neural network model. Figure 1C is not enough.

Response: We thank the reviewer's suggestion. In our revised manuscript, we added a detailed illustration of the neural network in **Figure S1**, which was reproduced below in **Figure R13**.

14. Does ECNet capable of predicting multiple functions? Is there any possibility?

Response: In this work, we focused on predicting a single function of a protein. In principle, it is possible to extend ECNet to predict multiple functions by generalizing the objective function to a multi-task prediction objective. In this case, the neural network will generate multiple outputs, where each output corresponds to a function. The overall loss function will be the (weighted) sum of individual loss for each

function prediction. As the testing is beyond the scope of this current work (e.g., experimentally measuring multi-function of protein variants), we leave it as a future direction.

15. Do the authors analyze what fraction of engineered cases retained the same function, and what fraction changes the function?

Response: In the context of our TEM-1 experiment, the first question from the reviewer is “what fraction of engineered cases *retain* the drug resistance?” Answering this question requires defining a fitness cutoff to differentiate resistant variants from non-resistant ones. In principle, any variant with a fitness (see Methods “Fitness calculation” for the definition) greater than 0 is resistant to ampicillin to some extent. Among the 37 engineered variants, 100%, 100%, and 94.6% variants have shown a non-zero fitness for ampicillin concentrations 300, 1500, and 3000 $\mu\text{g}/\text{mL}$, respectively.

We also try to answer a question stronger than the reviewer’s original question, i.e., “what fraction of engineered cases have a *stronger* drug resistance than the wildtype?” The data has been shown in **Figure 5c** in our manuscript: averaging over the six replicates, there are 51%, 91%, and 94% variants with higher resistance than the wildtype at ampicillin concentrations 300, 1500, and 3000 $\mu\text{g}/\text{mL}$, respectively.

The second question from the reviewer, in the context of our TEM-1 experiment, is “What fraction of variants gained a function *other than* drug resistance?” Unfortunately, as we mainly focused on ampicillin resistance in our work, we did not measure the levels of other functions for those variants. In addition, it is non-trivial to decide the possible functions those variants are likely to gain. Designing experiments to measure functions other than ampicillin resistance also requires additional experimental effort. Therefore, we think this analysis is beyond the immediate scope of our study.

16. Please provide more details on hyperparameter optimization of ECNet.

Response: We performed an inner-loop grid search using the training data of a target protein to decide the hyperparameters of ECNet. No test data was used to decide the hyperparameters. We have revised our manuscript as below to include the details on hyperparameter tuning of ECNet.

Revised text (Section “Methods - Training details”):

To select the hyperparameters of ECNet, we performed a small-scale grid search using the training data of a protein, such that $\frac{7}{8}$ of the training data was used to train a model with a specific set of hyperparameters, and the remaining $\frac{1}{8}$ data was used as the validation set to select the hyperparameters. The test set was not used for hyperparameter selection. In particular, we tested the LSTM’s dimension of $d_{LSTM} = 32, 64, \text{ and } 128$, the top layer dimension of $d_h = 32, 64, \text{ and } 128$, the reprojected embedding dimension of $d_{proj} = 128 \text{ and } 256$. In general, we found that $d_{LSTM} = 128$, $d_h = 128$, and $d_{proj} = 128$ are reasonably good defaults and can be used for a new protein.

Nevertheless, a careful grid search of hyperparameters for the new protein would further improve the model performance. Unless otherwise specified, the batch size was set to 128 and the maximum number of training epochs was set to 2,000 with an early stop if the performance has not been improved for 1000 epochs. Model training was performed on an Nvidia TITAN X GPU.

17. On page 23, “The batch size was set to 128 and the maximum number of training epochs was set to 1,000 with an early stop if the performance has not been improved for 500 epochs.” And in page 28, “The model was trained for 2,000 epochs with early stopping if its performance on the validation set did not improve for 1,000 epochs.” Are they two different models? How many different models are there? It will be good if authors provide detailed data (maybe tabular or so) describing how many models are there and which model uses what sort of features, data and comparing with which existing methods.

Response: Thanks for spotting our typos/inconsistency. Both models were trained for a maximum of 2,000 epochs with an early stop of 1000 epochs. We have fixed this typo in our manuscript. The two models here are conceptually the same. They should be thought of as two instantiations of ECNet, i.e., they share the same design choice, e.g., neural network architecture, training objective, input features, etc. The only difference between the two models is the training data (because they are trained to predict the function of different proteins).

Conceptually, there are three variants of ECNet used in our work. The primary one is the standard ECNet, including the two models mentioned above, which was used in all benchmarking experiments and performance analyses in our manuscript. The second variant of ECNet is the model used to engineer new TEM-1 mutants. In this model, we used an auxiliary classification loss to encourage the model to rank positive mutants (with fitness greater than the wild type) higher. The third variant of ECNet is an unsupervised model that does not require DMS data for training. We used this unsupervised ECNet model to demonstrate the utility of ECNet where DMS data is limited. All details of those ECNet models were described in the Methods section.

We also provide **Table R2** describing the characteristics of different ECNet models and other existing methods used in our manuscript.

Method	Supervised? (Model)	Objective	Input feature	Training data	Related experiments
ECNet (standard)	Supervised (LSTM)	Regression	Global & local evolutionary contexts, sequence	DMS	Benchmarking experiments and performance analyses
ECNet (TEM-1)	Supervised (LSTM)	Regression + Classification	Global & local evolutionary contexts, sequence	DMS of single-mutation and consecutive double-mutation	Engineering new TEM-1 variants

				TEM-1 variants	
ECNet (unsupervised)	Unsupervised (LSTM)	Language model objective (Maximize likelihood)	Sequences	Homologous sequences	Fitness prediction of viral proteins; Effects of training data size
Yang et al.	Supervised (Gaussian process)	Regression	Doc2Vec sequence embeddings	DMS	Benchmarking experiments
Envision	Supervised (Gradient boosting tree)	Regression	Biological, structural, and physicochemical features	DMS	Benchmarking experiments
EVmutation	Unsupervised (Markov random field)	Maximize likelihood	Sequences	Homologous sequences (MSA)	Benchmarking experiments
DeepSequence	Unsupervised (Sequence VAE)	Maximize likelihood	Sequences	Homologous sequences (MSA)	Benchmarking experiments
Autoregressive	Unsupervised (CNN)	Language model objective (Maximize likelihood)	Sequences	Homologous sequences	Benchmarking experiments
TAPE	Unsupervised (Transformer)	Language model objective (Maximize likelihood)	Sequences	Homologous sequences	Benchmarking experiments
UniRep	Unsupervised (LSTM)	Language model objective (Maximize likelihood)	Sequences	Homologous sequences	Benchmarking experiments
CSCS	Unsupervised (LSTM)	Language model objective (Maximize likelihood)	Sequences	Homologous sequences	Benchmarking experiments

Table R2. Characteristics of different ECNet models and other existing methods. ECNet and other algorithms are compared in terms of model type, training objective, input feature, training data, and their related experiments in this work. (DMS: deep mutational scanning; LSTM: long short-term memory network; MSA: multiple sequence alignment; VAE: variational autoencoders; CNN: convolutional neural network.)

Some minor suggestions to improve the readability of the manuscript-

18. Protein engineering involves mutation and InDel operations on protein. Although, the authors focused on mutation only and mostly focused on sequence-to-function mapping but “Protein

engineering“ as the title of the manuscript is a bit misleading. Authors may consider making it more specific.

Response: Thanks for the suggestion. We revised the Discussion section of our manuscript to make it more specific. We also reviewed existing studies that consider substitutions or insertion/deletion. The revised text is shown below.

Revised text (Section “Discussion”):

Supervised machine learning models have been explored recently to predict protein sequence-function relationships⁵²⁻⁵⁴. As in those studies, in this work, we mainly focused on improving the function by introducing point mutations, while introducing insertion/deletion was also explored in other work (Shin et al., 2021).

19. It is not clear from the main text regarding the measure of “16%-60%” improvement.

Response: The improvement is in terms of the spearman correlation. We have clarified this in our manuscript.

20. Page 27, first line – is there a typo? Please check.

Response: Thanks. We have fixed this typo.

21. It will be good if few lines can be added to emphasize more on the protein language model.

Response: In the Introduction section, we have added more information to emphasize more on the protein language model. The text is reproduced below.

Revised text (Section “Introduction”):

More recently, inspired by the advances in natural language processing, a trend is emerging that pre-trains a language model (LM) on large protein sequence datasets to learn the distribution of protein sequences^{13,16-22}. The protein sequences observed in nature today are the results of natural selection by evolution. Out of the possible mutations to a sequence, evolution samples those that preserve or improve the protein's fitness, such as stability, structure, and function. The underlying constraints or factors that determine protein's fitness have shaped the distribution of protein sequences. LMs are used to unravel the ‘grammars’ or ‘semantics’ of sequence generation by evolution. By being trained on natural sequences to predict the likelihood that a particular amino acid appears within a context, the language model learns representations that are semantically rich and encode structure, evolutionary and biophysical contexts¹⁶. It was found that using the learned

representation as the feature input to fine-tune a supervised model improves fitness prediction on multiple protein mutagenesis datasets¹⁷.

22. Please use the equation number wherever required. It will be convenient for the reader.

Response: Thanks for the suggestion. We have numbered all our equations in the Methods section.

Reviewers' Comments:

Reviewer #1:

None

Reviewer #2:

Remarks to the Author:

Thanks so much for putting together these really nice revisions; they strengthen the manuscript a lot.

In particular:

- The analysis of performance on noisy data is really interesting and nicely done, thank you for adding that. It highlights how brittle simple models like one-hot encoding are in comparison to learned models. In a similar vein, figure R8 shows how performance is impacted by the completeness of the DMS dataset. Both of these analyses are very important for experimentalists who want to reason about whether a particular assay/dataset may produce sufficiently clean data to work well with ML prediction methods. I hope more papers in this field will include this sort of analysis.
- The expanded analysis about how the size and quality of the MSA impacts performance is a nice addition. Figure R8 also dives into understanding what the model knows and how it works: it analyzes how the models' differential performance difference on highly conserved versus highly mutable residues.
- Thank you for clarifying how you selected hyperparameters.

It continues to be the case that in broad strokes this paper is awfully similar to quite a few papers that have come before it (improve prediction metric by a small amount; measure b-lactamase mutants). The authors claim that this is the first time that global evolutionary contexts have been combined with more specific information in the form of "local evolutionary contexts learned from an MSA". It is true this is the first time someone has published exactly this combination, although other papers have attempted to combine "global" and "local" information; i.e. "evotuning" of UniRep representations on related sequences.

That said, the new experiments added during revisions have made this a much stronger paper. The authors have done a very nice job testing the situations in which the model performs well, and using those results to understand how and why the model works. This is in itself a contribution; I would like to see more papers in the field like this.

Reviewer #3:

Remarks to the Author:

The authors' responses are satisfactory. The manuscript may be accepted.

Reviewer #2

Thanks so much for putting together these really nice revisions; they strengthen the manuscript a lot.

In particular:

- The analysis of performance on noisy data is really interesting and nicely done, thank you for adding that. It highlights how brittle simple models like one-hot encoding are in comparison to learned models. In a similar vein, figure R8 shows how performance is impacted by the completeness of the DMS dataset. Both of these analyses are very important for experimentalists who want to reason about whether a particular assay/dataset may produce sufficiently clean data to work well with ML prediction methods. I hope more papers in this field will include this sort of analysis.
- The expanded analysis about how the size and quality of the MSA impacts performance is a nice addition. Figure R8 also dives into understanding what the model knows and how it works: it analyzes how the models' differential performance difference on highly conserved versus highly mutable residues.
- Thank you for clarifying how you selected hyperparameters.

Response: We thank the reviewer for appreciating the new analyses in our revised manuscript.

It continues to be the case that in broad strokes this paper is awfully similar to quite a few papers that have come before it (improve prediction metric by a small amount; measure b-lactamase mutants). The authors claim that this is the first time that global evolutionary contexts have been combined with more specific information in the form of "local evolutionary contexts learned from an MSA". It is true this is the first time someone has published exactly this combination, although other papers have attempted to combine "global" and "local" information; i.e. "evotuning" of UniRep representations on related sequences.

That said, the new experiments added during revisions have made this a much stronger paper. The authors have done a very nice job testing the situations in which the model performs well, and using those results to understand how and why the model works. This is in itself a contribution; I would like to see more papers in the field like this.

Response: We thank the reviewer for the positive comments on the contribution of our work and the support of the acceptance of our manuscript.

Reviewer #3

The authors' responses are satisfactory. The manuscript may be accepted.

Response: We thank the reviewer for the positive feedback and the support of the acceptance of our manuscript.